# Active state structures of a bistable visual opsin bound to G proteins

Oliver Tejero[1,2], Filip Pamula[1,10], Mitsumasa Koyanagi[3,4], Takashi Nagata [5,11], Pavel Afanasyev [6], Ishita Das [7], Xavier Deupi [1,8,9], Mordechai Sheves[7], Akihisa Terakita [3,4], Gebhard F. X. Schertler [1] ✉, Matthew J. Rodrigues [1] ✉ & Ching-Ju Tsai [1] ✉

Opsins are G protein-coupled receptors (GPCRs) that have evolved to detect light stimuli and initiate intracellular signaling cascades. Their role as signal transducers is critical to light perception across the animal kingdom. Opsins covalently bind to the chromophore 11-cis retinal, which isomerizes to the all-trans isomer upon photon absorption, causing conformational changes that result in receptor activation. Monostable opsins, responsible for vision in vertebrates, release the chromophore after activation and must bind another retinal molecule to remain functional. In contrast, bistable opsins, responsible for non-visual light perception in vertebrates and for vision in invertebrates, absorb a second photon in the active state to return the chromophore and protein to the inactive state. Structures of bistable opsins in the activated state have proven elusive, limiting our understanding of how they function as bidirectional photoswitches. Here we present active state structures of a bistable opsin, jumping spider rhodopsin isoform-1 (JSR1), in complex with its downstream signaling partners, the $G_i$ and $G_q$ heterotrimers. These structures elucidate key differences in the activation mechanisms between monostable and bistable opsins, offering essential insights for the rational engineering of bistable opsins into diverse optogenetic tools to control G protein signaling pathways.

G protein-coupled receptors (GPCRs) play a pivotal role in cellular signaling by transducing extracellular signals into intracellular responses. Among the exceptionally diverse GPCR subtypes, light-sensitive opsins constitute a subclass of class A GPCRs that mediate visual and non-visual light perception across the animal kingdom[1]. Animal opsins achieve light sensitivity by covalent binding of the chromophore 11-cis retinal to a lysine side chain via a Schiff base. Photoisomerization of the bound retinal results in the all-trans isomer, which functions as an agonist and triggers protein conformational changes that lead to receptor activation. While vertebrate visual opsins bleach upon light activation by losing the chromophore due to hydrolysis of the Schiff base bond, retinal in invertebrate opsins and

[1]Laboratory of Biomolecular Research, PSI Center for Life Sciences, Villigen-PSI, Switzerland. [2]Department of Biology, ETH Zurich, Zurich, Switzerland. [3]Department of Biology, Graduate School of Science, Osaka Metropolitan University, Osaka, Japan. [4]The OMU Advanced Research Institute of Natural Science and Technology, Osaka Metropolitan University, Osaka, Japan. [5]Department of Biology and Geosciences, Graduate School of Science, Osaka City University, Osaka, Japan. [6]Cryo-EM Knowledge Hub, ETH Zurich, Zurich, Switzerland. [7]Department of Molecular Chemistry and Materials Science, Weizmann Institute of Science, Rehovot, Israel. [8]Condensed Matter Theory Group, Laboratory of Theoretical and Computational Physics, PSI Center for Scientific Computing, Theory and Data, Villigen-PSI, Switzerland. [9]Swiss Institute of Bioinformatics, Lausanne, Switzerland. [10]Present address: Department of Molecular Biology and Genetics, Aarhus University, Aarhus, Denmark. [11]Present address: Institute for Solid State Physics, The University of Tokyo, Kashiwa, Chiba, Japan. ✉e-mail: gebhard.schertler@psi.ch; matthew.rodrigues@psi.ch; ching-ju.tsai@psi.ch

non-visual vertebrate receptors remain stably bound and may photo-isomerize back to the 11-cis form. Vertebrate visual opsins are therefore referred to as "monostable", while invertebrate and non-visual opsins are termed "bistable"[1–3]. Monostable opsins bind $G_{i/o/t}$ subtypes while bistable opsins primarily bind the $G_q$ subtype, but also $G_{i/s/o}$ subtypes[3].

As invertebrates have evolved to occupy niches in very diverse terrestrial, marine, subterranean, and freshwater environments, their visual systems have adapted to the needs of their habitats. As such, invertebrate opsins have evolved to absorb light from the ultraviolet (330 nm) to the red (600 nm) range of the spectrum depending on the needs of the organism[4]. In addition to their role in invertebrate vision, bistable opsins are essential for non-visual light perception in vertebrates, regulating processes including photoentrainment of circadian rhythms, thermogenesis, and vascular development in mammals[1–3,5–8].

Formation of a protonated Schiff base (PSB) with the opsin shifts the absorption maximum of retinal from the ultraviolet (~ 370 nm) to the visible range of the spectrum (440 nm – 700 nm). The positively charged Schiff base must be stabilized by a nearby negatively charged amino acid residue termed the counterion. Nearly all opsins feature $Glu^{45.44}$ (GPCR general numbering scheme[9]) at extracellular loop 2 (ECL2), referred to as 'ancestral' or 'distal', which in bistable opsins plays the role of the counterion[10–12]. In addition, monostable opsins in vertebrates have evolved a proximal $Glu^{3.28}$ at the transmembrane helix 3 (TM3), which performs the role of counterion in the inactive state[13,14]. This position is occupied by a conserved tyrosine ($Tyr^{3.28}$) in invertebrate bistable opsins[10–12]. In all opsins, the shape of the tight retinal binding site results in severe steric constraints on its conformation. Polar and charged nearby amino acids influence the distribution of the positive charge of the protonated Schiff base along the chromophore polyene chain in the ground and excited states, and thereby the energy required to electronically excite the retinal. Understanding the nature of the retinal binding site is, therefore, key to determining how opsins have evolved to absorb specific wavelengths of light with high sensitivity and control of the isomerization product.

Jumping spider rhodopsin isoform-1 (JSR1) is an invertebrate bistable opsin essential for depth perception in jumping spiders[15]. It absorbs maximally at 535 nm in the inactive state bound to 11-cis retinal[15,16]. Photo-isomerization of retinal from the 11-cis isomer to the all-trans configuration leads to the formation of an active receptor conformation capable of interacting with intracellular G proteins[11,16]. We have shown that JSR1 can couple to and activate human $G_i$ and $G_{q/11}$ proteins in vitro[11,17,18]. As a bistable opsin, retinal in JSR1 undergoes a reversible cis-trans/trans-cis photo-isomerization[1,2,16,19]. The ability to maintain the Schiff base in a protonated state throughout the photocycle ensures thermal stability of the Schiff base in the active state and allows retinal to revert to the 11-cis form after a second photo-isomerization event, thereby conferring bistability to the receptor[10,16]. JSR1 bound to 9-cis retinal has a blue-shifted absorption maximum ($\lambda_{max} = 505$ nm) compared to JSR1 bound to the native 11-cis retinal ($\lambda_{max} = 535$ nm)[15,16]. Such $\lambda_{max}$ separation allows identification and monitoring of the dark (using 9-cis retinal) and illuminated (all-trans-retinal) states. We have also shown that for both 9- and 11-cis retinal, photo-isomerization generates activated JSR1 bound to all-trans-retinal with a $\lambda_{max}$ of 535 nm[15,16].

High-resolution crystal structures of invertebrate JSR1 and squid rhodopsin have shed light on the architecture of their retinal-binding sites and inactive state conformations[11,20,21]. In addition, crystal structures of the batho and lumi intermediate states of squid rhodopsin provided the first insights into the retinal isomerization process[22,23]. However, critical aspects of their active-state conformation and interactions with downstream signaling partners remain unexplored. For instance, the lack of structural data on signaling complexes of bistable opsins has hindered the efforts to determine their activation

mechanism and how they may revert back to the 11-cis-bound inactive state.

Here, we present cryo-EM structures of the active state of JSR1 in complex with $G_i$ and $G_q$ signaling partners. The structures provide detailed insights into the molecular architecture and conformational changes that underlie the activation of bistable opsins and their interaction with G proteins. We characterize the retinal binding pocket and compare it to that of the monostable bovine rhodopsin. In addition, we compare the conserved microswitch domains between JSR1 and prototypical class A GPCRs to identify similarities and differences in the activation mechanisms of the receptors.

This study lays the groundwork for future investigations into the activation dynamics of these unique light-sensitive receptors, offering alternative avenues for the development of bistable opsins as optogenetic tools in GPCR-based signaling research.

## Results

To obtain active JSR1 for forming JSR1-G protein complexes, we first produced apo-form JSR1 by expressing the wild-type receptor in HEK293 cells lacking *N*-acetylglucosaminyltransferase I, followed by reconstituting JSR1 with the endogenous agonist, all-trans-retinal. However, it was not possible to identify conditions under which all-trans retinal would bind the receptor[18]. We, therefore, adopted two alternative strategies: (i) reconstitution of inactive JSR1 with 9-cis retinal and generation of the active form by illumination, and (ii) reconstitution of active JSR1 with the non-natural agonist all-trans retinal 6.11 (ATR6.11) (Fig. 1a)[18].

Active JSR1 was reported to couple to the human $G_i$ (h$G_i$) and human $G_q$ (h$G_q$) heterotrimers in vitro[18]. When we attempted to form a JSR-G protein complex, the h$G_i$ heterotrimer showed favorable biochemical characteristics compared to the h$G_q$ and was therefore used to form the complex. The JSR1•all-trans retinal-human $G_i$ complex (JSR1-h$G_i$) was prepared by reconstituting JSR1 with 9-cis retinal and illuminating with 495 nm long-pass filtered light under light-controlled conditions to activate the receptor and induce coupling to the h$G_i$ heterotrimer. A sub-5 Å resolution map was obtained from this JSR1•all-trans retinal-h$G_i$ complex (Supplementary Figs. 1a, b, 2b, 3 and 4 and Supplementary Table 1). Several obstacles prevented the determination of a higher-resolution structure. Firstly, steady state illumination of JSR1 generates a dynamic equilibrium of retinal in the 9-cis, 11-cis, and all-trans isomers with $\lambda_{max}$ within 30 nm of each other, producing a mixed population of active/inactive states due to back-isomerization[16]. Secondly, this JSR1-h$G_i$ complex was prone to dissociation during grid preparation, suggesting a low affinity between the receptor and the G protein heterotrimer; this also reduced the number of fully assembled complex particles in the EM dataset. Finally, the flexibility of the JSR1-h$G_i$ complex resulted in heterogeneity in the particle set, lowering the overall resolution of the cryo-EM map. Nevertheless, a structural model built based on this map still provides valuable insights (Supplementary Figs. 2b and 4).

We, therefore, adopted the second strategy mentioned above to prepare active JSR1. Rather than activating JSR1•9-cis retinal with light, we reconstituted JSR1 with ATR6.11, a non-natural retinal analog that has an agonist activity[18]. In addition, we attempted to use the jumping spider visual $G_q$ (js$G_q$) to form the complex. However, the production of active and pure js$G_q$ for structural studies was not successful. Therefore, we decided to create a human $G_i$/jumping spider $G_q$ chimera (js$G_{iq}$) by engineering h$G_i$ to match the sequence of js$G_q$ at the interface where the h$G_i$ protein contacts the receptor based on our JSR1-h$G_i$ cryo-EM structure. These residues, mainly in the C-terminal α5 helix, were substituted (Supplementary Fig. 5; see "Methods" for details). An in vitro activity assay showed that this chimera retains intrinsic GTPase activity (Fig. 1b). Moreover, JSR1 bound to ATR6.11 triggers an increase in GTPase activity, confirming that the receptor is

activated and can catalyze nucleotide exchange in jsG$_{iq}$, similarly to hG$_i$ (Fig. 1b).

The JSR1•ATR6.11-jsG$_{iq}$ complex (JSR1-jsG$_{iq}$) sample for the cryo-EM study was purified in the detergent dodecyl-β-D-maltoside (DDM) (Supplementary Fig. 1c and d). 3D classification and 3D variability single-particle analysis revealed conformational heterogeneity in the cryo-EM particle set; the Gα subunit shows a particularly high degree of flexibility that lowers the resolution in this area and, therefore, also the global resolution (Supplementary Fig. 6 and Supplementary Movies 1 and 2). In addition, there are larger motions of the G protein heterotrimer relative to JSR1. With 3D classification, cryo-EM maps for two conformations of the JSR1-jsG$_{iq}$ complex were separated and independently refined (Supplementary Fig. 6 and Supplementary Table 1). One map shows a fully engaged G protein similar to other class A GPCR-G protein complexes (Fig. 1c and Supplementary Fig. 6d), whereas the other shows a small rotation of the G protein with a somewhat different binding pose of the α5 helix (Supplementary Figs. 2a, 6e and Supplementary Movie 2). The map of this fully engaged G protein complex was refined to 4.1 Å (JSR1-jsG$_{iq}$_1), and the map with the rotated G protein was refined to 4.2 Å (JSR1-jsG$_{iq}$_2). Density for the α-helical domain (AHD) was observed in both JSR1-jsG$_{iq}$ maps, allowing rigid body fitting of the AHD into the map, but the resolution did not allow precise modeling of side chains. As the structures of the JSR1-jsG$_{iq}$_1 and JSR1-jsG$_{iq}$_2 retinal binding sites are highly similar, we focus on the JSR1-jsG$_{iq}$_1 structure due to its higher resolution. We compare the JSR1-jsG$_{iq}$_1 structure to those of JSR1-hGi and JSR1-jsG$_{iq}$_2 at the G protein binding site, where structural differences are observed.

## Overall structure of the JSR1-G protein complex

The overall structure of the JSR1-jsG$_{iq}$_1 complex shows the typical features of a class A GPCR-G protein complex (Fig. 1d). The outward movement of TM5 and TM6 relative to the inactive state opens a cytoplasmic cleft that allows jsG$_{iq}$ to bind to JSR1 (Fig. 1e). The C-terminus of jsGα$_{iq}$ forms a hook-like structure (C-hook) that inserts into the open binding cleft as observed in all the GPCR-G protein complexes (Fig. 1d). In contrast to other GPCR structures, we observe an elongated TM5 that interacts with the G protein. The local resolution of the intracellular loop 3 (ICL3) and the cytoplasmic end of TM6 is lower compared to the rest of the receptor, indicating greater flexibility in this area (Supplementary Fig. 6d and Supplementary Movie 1). Because of this, four residues at the start of TM6 (Asp265$^{6.23}$-Lys268$^{6.26}$) could not be modeled into the experimental map with confidence and were, therefore, omitted from the atomic model.

The N-terminus and extracellular loop 2 (ECL2) of JSR1 form a cap over the orthosteric retinal binding site, shielding it from the extracellular milieu. This cap is conserved in all opsins, but the structure of the cap can vary between receptors[11,13,20,24], indicating that it might be important for the stability of the retinal-binding pocket and receptor stability overall.

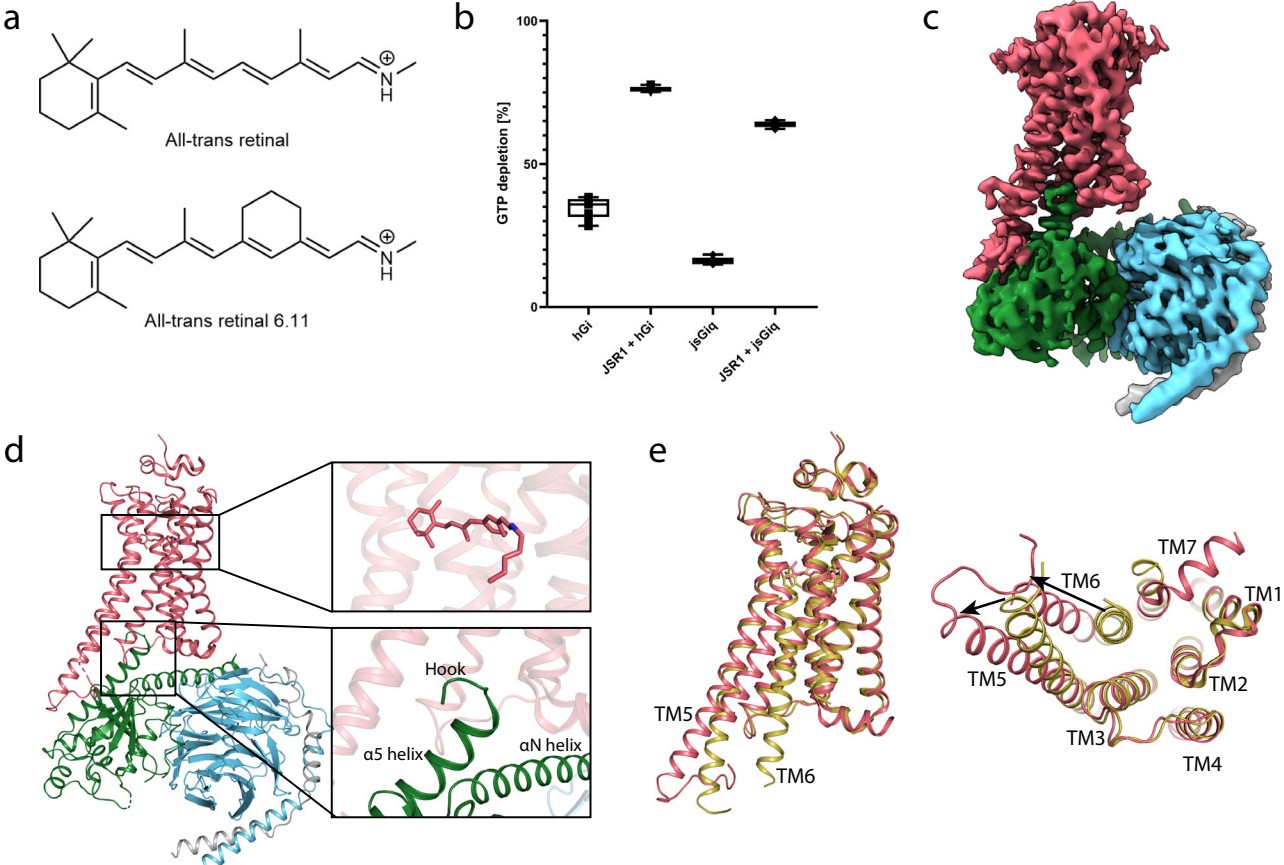

**Fig. 1 | Structural analysis of active-state JSR1 bound with all-trans-retinal analog ATR6.11 and G protein. a** Chemical structures of all-trans-retinal and the synthetic analog all-trans retinal 6.11, both with a protonated Schiff base linkage. **b** GTPase-Glo assay showing GTP depletion for hG$_i$ and jsG$_{iq}$ samples in the absence and presence of unilluminated JSR1 reconstituted with ATR6.11. Each condition was measured in octuplicates. Data is presented as mean values +/− standard deviation. **c** Cryo-EM map of JSR1-jsG$_{iq}$_1 colored by subunits. JSR1 in salmon, jsGα$_{iq}$ in green, Gβ in cyan, and Gγ in gray. **d** Overall model of the JSR1-jsG$_{iq}$_1 complex colored as described for the cryo-EM map. The retinal binding site and the G protein binding site are shown enlarged in the inset boxes. **e** Overlay of the structures of inactive JSR1•9-cis retinal (olive, PDB 6I9K) and active JSR1•ATR6.11 of the JSR1-jsG$_{iq}$_1 complex (salmon). The left panel shows a side view and the right panel shows a view from the cytoplasmic side of the receptor.

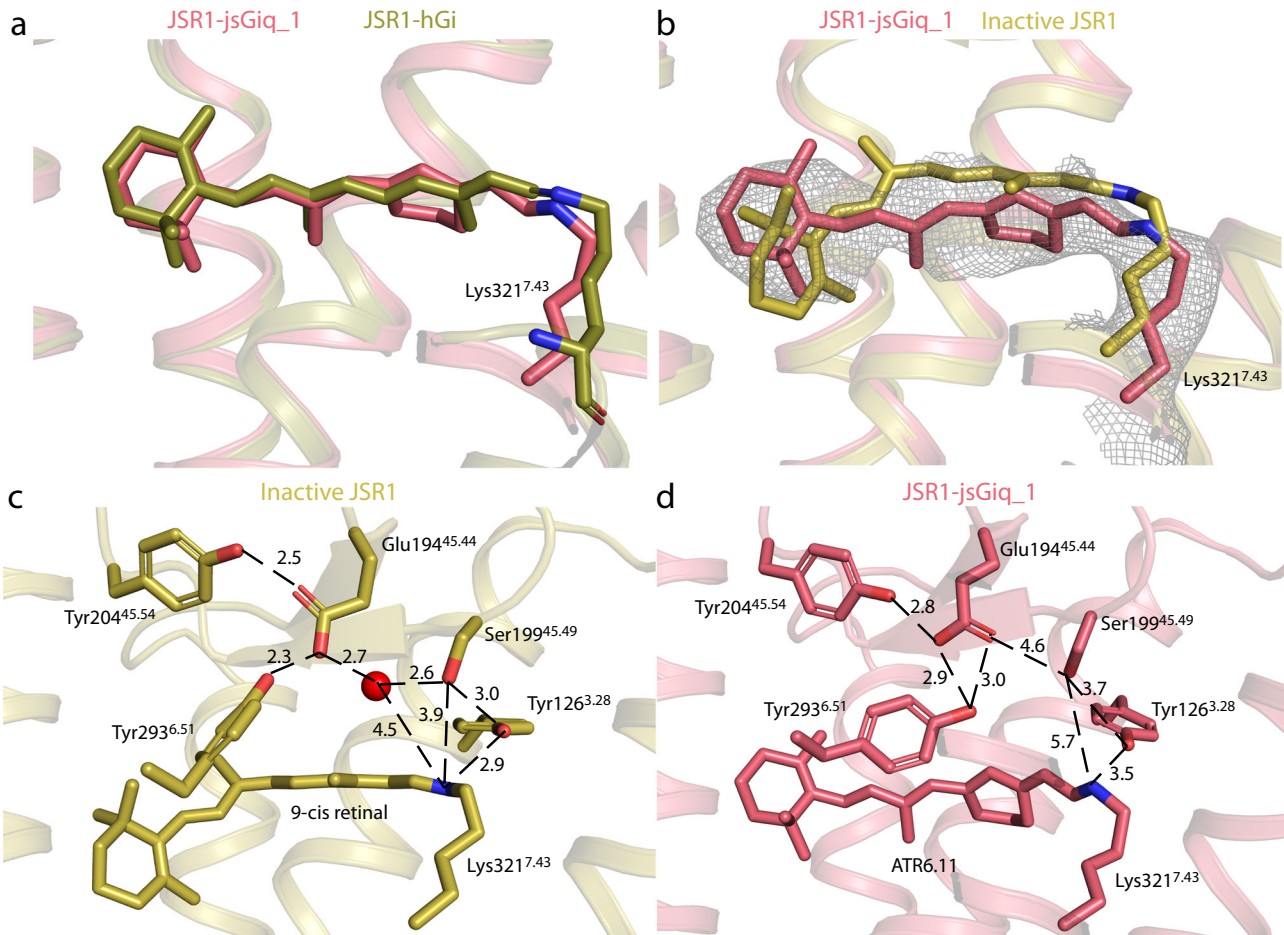

**Fig. 2 | Comparison of the retinal binding site in the inactive and active states of JSR1. a** Comparison of all-trans-retinal (JSR1-hG$_i$ complex, olive) and all-trans-retinal 6.11 (JSR1-jsG$_{iq}$_1 complex, salmon). In all the panels, carbon atoms are colored as in the corresponding JSR1 structure, with nitrogen atoms in blue and oxygen atoms in red. **b** Comparison of 9-cis retinal in the inactive JSR1 (PDB 6I9 K, olive) and all-trans-retinal 6.11 in the active JSR1 (salmon). Lys321[7.43], the Schiff base link, and the retinal ligand are shown as sticks. The cryo-EM map is shown for all-trans retinal 6.11. **c** Environment of 9-cis retinal in inactive JSR1. Residues involved in the Schiff base-counterion link are shown as sticks. **d** Environment of all-trans-retinal 6.11 in active JSR1. The same residues as in C are shown as sticks.

## Retinal binding pocket

The electron density in the orthosteric binding pocket shows the presence of ATR6.11 covalently linked to Lys321[7.43] via a PSB (Supplementary Fig. 9a and b). The JSR1-hG$_i$ map shows density for all-trans-retinal and overlays well with the ATR6.11 analog, indicating a similar binding pose for both ligands (Fig. 2a and Supplementary Fig. 9c and d). While the retinal poses are nearly identical in these two structures, there are small differences in the position of the lysine side chain and the Schiff base. These structural differences may reflect the ambiguity of modeling flexible side chain conformations at this resolution, here we focus on the JSR1-jsG$_{iq}$_1 structure, which was solved at a higher resolution and for which the side chain and retinal conformations could be modeled with greater confidence.

The density for the β-ionone ring of ATR6.11 is not sufficiently well defined to determine its orientation based on the experimental cryo-EM map alone. We have therefore modeled it rotated 180° relative to that of the 9-cis retinal β-ionone ring, which is consistent with the pose observed for all-trans-retinal in the active bovine rhodopsin metarhodopsin-II structure[24–27]. Based on the JSR1-jsG$_{iq}$_1 cryo-EM map, we observe that the polyene chain and Schiff base of ATR6.11 are shifted towards the intracellular side relative to the inactive 9-cis retinal form (Fig. 2b)[11]. Conversely, the β-ionone ring of ATR6.11 is shifted towards the extracellular side of the receptor (Fig. 2b)[11]. This pose of ATR6.11 agrees with that of all-trans-retinal in the active metarhodopsin-II state of bovine rhodopsin[24–27] (Supplementary Fig. 10c).

Comparison of the environments of retinal between the active and inactive states of JSR1 shows that 9-cis retinal is tightly packed in the inactive state with 15 residues within 4 Å (Supplementary Movie 3)[11], while ATR6.11 has only seven (Supplementary Movie 4). Similarly, 11-cis retinal in bovine rhodopsin is also tightly packed by surrounding residues with 15 residues within 4 Å (Supplementary Movie 5)[13], while the all-trans retinal only has eight residues (Supplementary Movie 6)[24]. This indicates that in the active state, water molecules surround the retinal in the retinal binding pocket and bridge interactions with nearby side chains.

In contrast to the monostable bovine rhodopsin, the Schiff base in JSR1 is not deprotonated and hydrolyzed upon retinal isomerization to the all-trans conformation[2,12]. The proximal counterion in monostable opsins, Glu[3.28], together with a water molecule, is partially responsible for the hydrolysis of the Schiff base[27–29]. In JSR1, this position is occupied by a bulkier Tyr126[3.28], leaving no space for a water molecule at a similar position as is seen in the active bovine rhodopsin structures (Fig. 2d and Supplementary Fig. 10a, b)[27]. The presence of a tyrosine instead of glutamate at this position might contribute to the thermal stability of the Schiff base in the active state JSR1 and other invertebrate opsins. The inactive state crystal structure of JSR1 showed that the distal counterion Glu194[45.44] is too distant to interact directly and is instead linked to the PSB via a water-mediated hydrogen bond network (Fig. 2c)[11]. The JSR1 retinal binding pocket undergoes subtle changes from the inactive to the active state. The movement of the polyene

chain and Schiff base towards the intracellular side increases the distance to Tyr126[3.28] (2.9 Å to 3.5 Å) and Ser199[45.49] (3.9 Å to 5.7 Å) (Fig. 2c, d and Supplementary Fig. 9e)[11]. In a previous work[12], we have shown that mutating Tyr126[3.28] to phenylalanine results in no change in the Schiff base pKa, indicating that the hydroxyl group of Tyr126[3.28] does not influence the protonation state of the Schiff base in either active or inactive states. The distal counterion, Glu194[45.44], also changes its rotamer in the active state but is still oriented by the "glutamate cage" formed by Tyr204[45.54] and Tyr293[6.51] as observed in the inactive state (Fig. 2c, d)[11]. In our structure, the proton of the PSB is oriented towards Glu194[45.44] but is not close enough to interact directly with the distal counterion (Fig. 2d). Ser199[45.49] adopts a different conformation compared to the inactive state and orients away from the PSB, implicating that Ser199[45.49] is not directly involved in the PSB-counterion link in the active state. These observations are consistent with published mutagenesis data suggesting that Glu194[45.44] acts as the counterion in the active state, but that Ser199[45.49] is not involved in the hydrogen bond network linking the PSB to the Glu194[45.44] [12].

In the crystal structure of the inactive state, ordered water molecules form key polar interactions linking the PSB to the counterion[11]. In our active structure, there is sufficient space in the binding pocket for water molecules to mediate a similar hydrogen bond network. Interestingly, the experimental cryo-EM map shows a patch of density between the counterion and the PSB (Supplementary Fig. 9f). This density may be attributed to one or two ordered water molecule(s) that mediate polar interactions between the PSB and the counterion, however further experimental evidence is required to confirm this observation.

## Micro-switch domains

Retinal isomerization is the first step in the activation process that triggers conformational changes in the protein leading to the active state of JSR1. In class A GPCRs, including JSR1, there are several known conformational changes in conserved micro-domains required for receptor activation. In this section, we investigate four of these motifs and compare them to the monostable bovine rhodopsin. These motifs are the C-W-x-P motif in TM6, the P-I-F motif between TM3/5 and 6, the D/E-R-Y motif in TM3, and the N-P-x-x-Y motif in TM7.

The C[6.47]-W[6.48]-x-P[6.50] motif is highly conserved in class A GPCRs, but invertebrate opsins feature an alanine or serine at position 6.47 rather than cysteine (Supplementary Fig. 12c). Pro[6.50] is conserved in the opsin family and decisively shapes TM6. Interestingly, in most opsins, the proline is followed by a conserved Tyr[6.51] that is implicated in the coordination of the distal counterion (Fig. 2c, d and Supplementary Figs. 10a, b and 12c)[11,13,27]. Trp[6.48] is positioned just below the retinal binding site and is one of the major activation micro-switches. The polyene chain of all-trans retinal pushes on Trp[6.48] resulting in a cascade of structural changes that lead to the outward swing of TM6 (Fig. 3a and Supplementary Figs. 11a and 12a)[30,31]. Trp[6.48] mainly shifts towards the intracellular side of the receptor, similar to bovine rhodopsin (Fig. 3a, b and Supplementary Figs. 11a, 12a, b)[11,13,27]. This shift initiates the outward movement of TM6 and subsequent conformational changes of the residues in TM6.

One helix turn below Trp[6.48] lays Trp[6.44], part of the P[5.50]-I[3.40]-F[6.44] motif and highly conserved in class A GPCRs. In this motif, JSR1 features a tryptophan instead of a phenylalanine at position 6.44. The analysis of sequence alignments shows that while vertebrate opsins almost exclusively have the expected phenylalanine, invertebrate opsins mainly feature a tryptophan, with only a few phenylalanine or tyrosine residues (Supplementary Fig. 12d). The position of the tryptophan in JSR1 is highly similar to the position of the equivalent phenylalanine in bovine rhodopsin in both the active and inactive states (Fig. 3c and d)[11,13,27]. However, in JSR1, the translational shift of P[5.50] is smaller, and I[3.40] does not undergo a rotamer change but only a small

translational shift relative to the inactive state (Fig. 3c, d and Supplementary Fig. 11b)[11,13,27].

The conserved D/E[3.49]-R[3.50]-Y[3.51] motif (Supplementary Fig. 12e) is crucial to stabilize the active state of the receptor and its interactions with the G protein. In the inactive state of JSR1, the DRY motif is in a closed conformation where Arg148[3.50] forms an intra-helical ionic interaction with Asp147[3.49] and an inter-helical ionic interaction with Glu272[6.30] (Fig. 3e, f). In the active state of JSR1, both ionic interactions are broken, and Arg148[3.50] adopts an extended conformation that interacts with the G protein (Fig. 4b and Supplementary Table 2). The rotamer change of Arg148[3.50] is further stabilized by interactions with Tyr234[5.58] and Tyr331[7.53] (Fig. 4). This is very similar to, for instance, the prototypical class A GPCRs bovine rhodopsin and human β2-adrenergic receptor, in which it was shown that this interaction is critical in forming the active state[32,33]. In bovine rhodopsin, the corresponding Tyr223[5.58] points away from the transmembrane core in the inactive state and moves toward the transmembrane core in the active state[32,33]. In contrast, in the inactive state of JSR1, this Tyr[5.58] is already oriented towards the core and only changes its rotamer in the active state, more closely resembling the β2-adrenergic receptor, a prototypical class A GPCR binding diffusible ligands (Fig. 4 and Supplementary Fig. 13)[11,34,35]. Thus, the sequence and local structure of the D/E[3.49]-R[3.50]-Y[3.51] motif and surrounding residues of the bistable JSR1 are more similar to those of non-photosensitive class A GPCRs than to the monostable light-sensitive bovine rhodopsin.

In JSR1-jsG$_{iq}$_1, as in other class A GPCRs[36,37], TM7 relocates towards the transmembrane core upon activation, and Tyr331[7.53] of the conserved N[7.49]-P[7.50]-x-x-Y[7.53] motif shifts into the core to pack against Leu141[3.44] and Ile144[3.47] close to Tyr234[5.58] and Arg148[3.50]. This arrangement allows Arg148[3.50] to interact with the backbone oxygen of Cys351[H5.23] (the superscript corresponds to the G protein general residue number[38]) of the C-hook in jsGα$_{iq}$ (Fig. 4 and see Supplementary Table 2 for detailed interactions) and stabilize the active complex. At the TM6/TM7 interface, we observe that the packing at Cys285[6.43] and Tyr326[7.48] appears to restrict the outward movement of TM6 (Supplementary Fig. 18, **top**). In the JSR1-hG$_i$ complex, a stronger packing leads to a more confined cytoplasmic opening that is just large enough to accommodate the small Gly352[H5.24] in the C-hook of Gα$_i$ near the DRY motif in TM3 and the TM7/helix 8 (H8) turn. Remarkably, this packing is looser in the two conformations of the JSR1-jsG$_{iq}$ complex (Supplementary Fig. 18, **bottom**), allowing a further relocation of TM6 to accommodate the larger Asn352[G.H5.25] in the C-hook of jsGα$_{iq}$. Subtle differences in the structure of TM7 between the two conformations result in slightly different but effective binding poses of the jsG$_q$ C-hook.

The conformational changes in the activation microswitches connecting the retinal binding site and the G protein binding site are summarized in Fig. 4. Together, these conformational changes lead to the opening of a cytoplasmic cleft that allows the G protein to be activated by the receptor.

## G protein binding pose and conformation

Formation of the cytoplasmic cleft in the activated receptor enables binding of the G protein, which is a prerequisite for catalysis of nucleotide exchange by the activated receptor. The binding poses of Gα to the receptor are overall similar in the JSR1-jsG$_{iq}$_1, JSR1-jsG$_{iq}$_2, and JSR1-hG$_i$ structures. Still, we observe minor but noticeable differences in the orientations of the whole G protein heterotrimer relative to the receptor, as well as small differences in the cytoplasmic cleft opening (Fig. 5).

As the ionic lock at the D/E[3.49]-R[3.50]-Y[3.51] motif breaks upon activation of the receptor, Arg148[3.50] forms direct interactions with the C-hook of both hGα$_i$ and jsGα$_{iq}$ (Supplementary Tables 2 and 3). This interaction is universal in class A GPCR-G protein complexes regardless of the G protein subtype. However, we observe that the precise nature

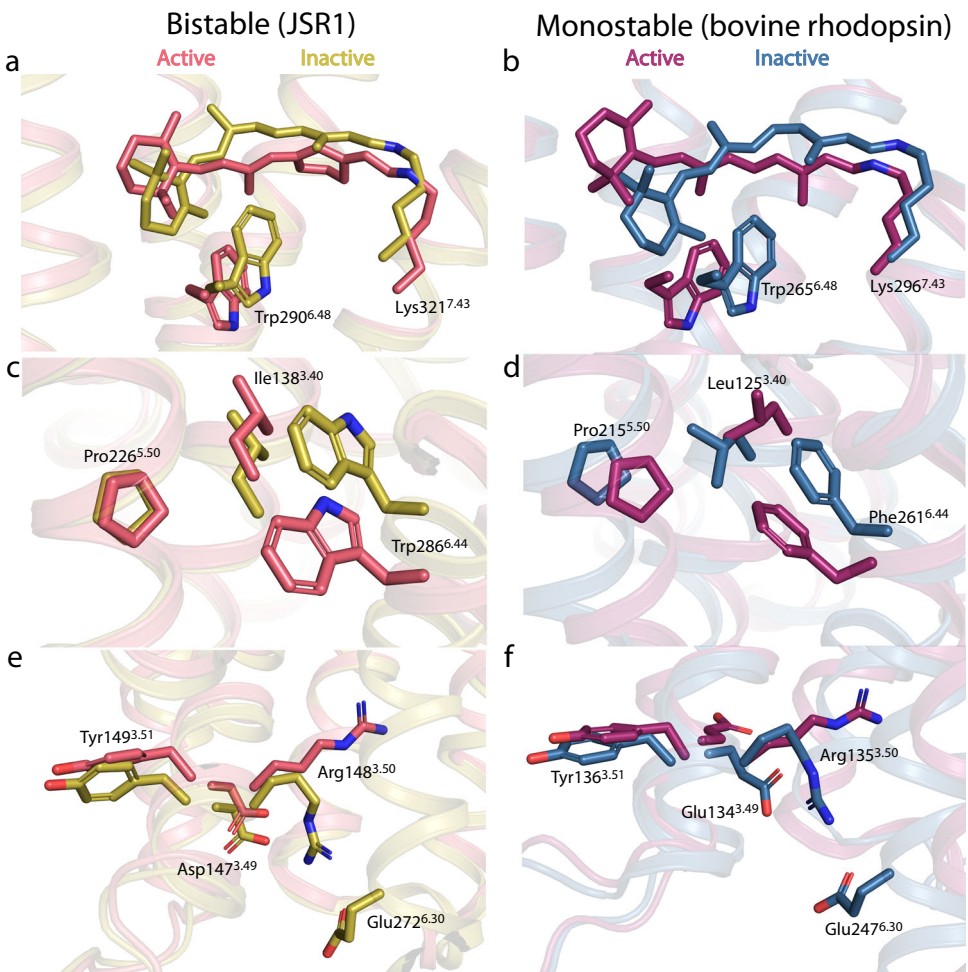

**Fig. 3 | Comparison of the micro-switch domains between inactive and active JSR1, and between inactive and active bovine rhodopsin. a** Comparison of the retinal binding site between the inactive (olive) and active (salmon) states of JSR1. The C-W-x-P motif Trp290[6.48], retinal, and the connecting Lys321[7.43] are shown as sticks. **b** Comparison of the inactive (blue) and active (purple) states of bovine rhodopsin, showing retinal Lys296[7.43], and the conserved Trp265[6.48] as sticks. **c** Comparison of the P-I-F motif (shown as sticks) between the inactive and active states of JSR1. **d** Comparison of the P-I-F motif (shown as sticks) between the inactive and active states of bovine rhodopsin. **e** Comparison of the D-R-Y motif

(shown as sticks) between the active and inactive states of JSR1. Glu272[6.30], which forms interaction with Arg147[3.49] in the inactive state, is also shown. **f** Comparison of the E-R-Y motif (shown as sticks) between the inactive bovine rhodopsin with metarhodopsin-II. Glu247[6.30] from the inactive state is also shown. Compared structures are inactive JSR1 (PDB 6I9K, olive), active JSR1 (JSR1-jsG$_{iq}$_1 from this work, salmon), inactive bovine rhodopsin (PDB 1GZM, blue), and active metarhodopsin-II (PDB 5EN0, purple). In all panels, carbon atoms are shown in the respective color of the receptor, nitrogen atoms in blue, and oxygen atoms in red.

of the interaction between Arg148[3.50] and Gα varies depending on the specific binding pose of the C-hook. Specifically, Arg148[3.50] engages to form more interactions with the C-hook in JSR1-G$_{iq}$_1 than in JSR1-G$_{iq}$_2 (Supplementary Tables 2 and 3). In addition to this interaction, in all three structures, JSR1 also contacts Gα at TM5/ICL3/TM6 and at the TM7/H8 turn, but with differences in the degree of opening of the cytoplasmic cleft. In the two conformations of the jsG$_{iq}$ complex, the outward movement of TM5 and TM6 is larger than in the hG$_i$ complex, resulting in a somewhat larger space between TM5, TM6, and TM7 that allows a deeper insertion of the jsGα$_{iq}$ α5 helix. In contrast, a smaller space in the hGα$_i$ complex is associated with a shallower binding of the C-hook (Fig. 5a, b and Supplementary Fig. 14a). This shallow binding also echoes the dissociation of the complex that we observed in the JSR1-hG$_i$ cryo-EM data, as both phenomena point toward a low affinity between JSR1 and hG$_i$. Overall, our JSR-G protein structures agree well with previous observations that the magnitude of the outward shift of TM6 largely depends on the subtype of Gα protein, especially the sequence of the α5 helix (Supplementary Fig. 14)[39,40].

While the cytoplasmic opening of the receptor is similar in both jsG$_{iq}$-bound structures, in JSR1-jsG$_{iq}$_2, TM6 and TM7 are shifted

together ~2 Å further towards the direction of ICL1 to accommodate the slightly rotated pose of jsGα$_{iq}$ (Fig. 5d). This is apparently linked to an interaction between TM5/ICL3 and jsGα$_{iq}$. Residues R261-N263 of ICL3 are resolved in JSR1-jsG$_{iq}$_1 due to their interaction with the α4 helix/β6 strand of jsGα$_{iq}$; these residues are however not resolved in JSR1-jsG$_{iq}$_2 (Supplementary Fig. 15 and Supplementary Tables 2 and 3), suggesting that this part of ICL3 is structurally flexible and interacts transiently with Gα. We hypothesize that in JSR1-jsG$_{iq}$_2, loosening this interaction mediated by ICL3 results in a different accommodation of the entire jsGα$_{iq}$, which can be measured as a 17° rotation of the α5 helix. Interestingly, in JSR1-hG$_i$, we can observe clear density for the entire ICL3 (Supplementary Fig. 15a), showing how this flexible region can be stabilized by interaction with the G protein, as we also observe in JSR1-jsG$_{iq}$_1. Furthermore, both JSR1-jsG$_{iq}$ models are incomplete at the cytoplasmic end of TM6, indicating that this region might not form an α-helix but rather remain as a flexible loop-like structure in the active state. The JSR1-hG$_i$ structure shows an atypical bend of TM6 exactly where the density disappears in the JSR1-jsG$_{iq}$ maps, suggesting that the α-helical structure might also be disrupted in this complex (Supplementary Fig. 15).

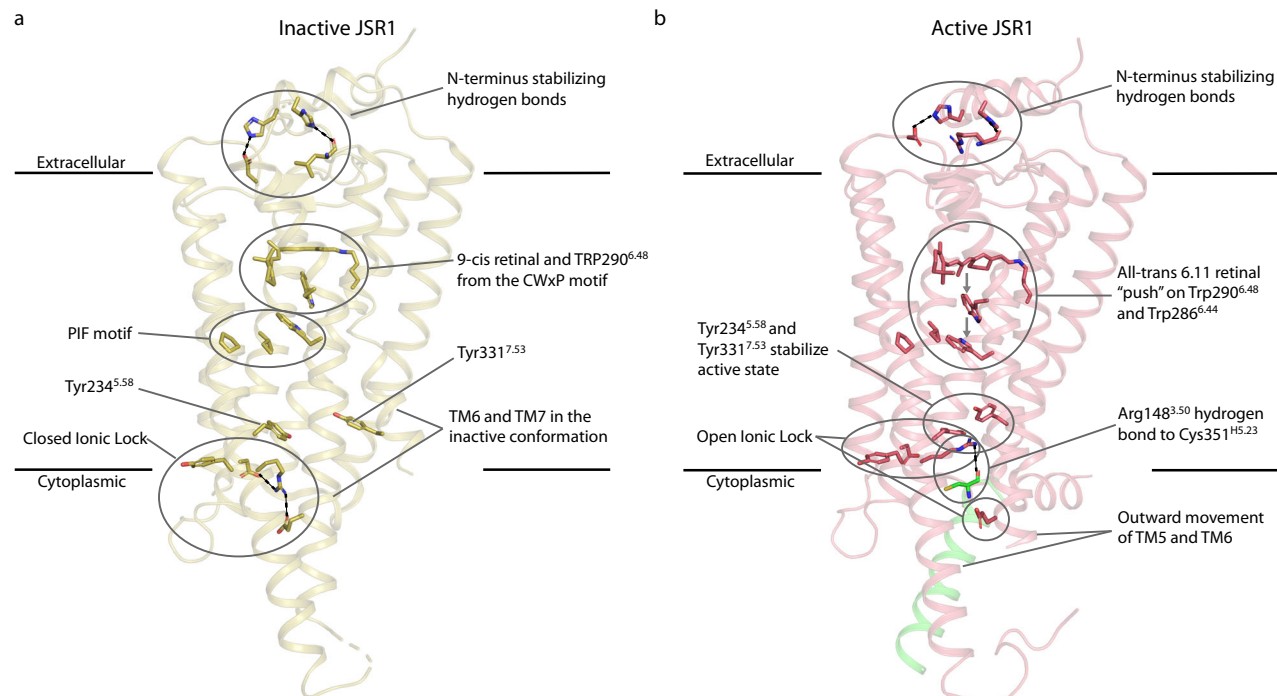

**Fig. 4 | Activation of JSR1 by the agonist all-trans 6.11 retinal. a** Structure of inactive JSR1 (PDB 6I9K) is shown in olive. Side chains of the key residues are shown as sticks. **b** Structure of JSR1-jsG$_{iq}$_1. The receptor is shown in salmon and the C-terminus (residues 344–354) of jsGα$_{iq}$ in green. The side chains of key residues involved in activation are shown as sticks.

The density of the terminal C-hook is better resolved in JSR1-jsG$_{iq}$_1 than in JSR1-jsG$_{iq}$_2, indicating that this region is more flexible/heterogeneous in the JSR1-jsG$_{iq}$_2 conformation. The C-hook also engages in fewer contacts in the JSR1-jsG$_{iq}$_2 conformation (five interactions) compared to JSR1-jsG$_{iq}$_1 (eight interactions) (Supplementary Tables 2 and 3). There are also fewer total interactions between the receptor and the Gα subunit in JSR1-jsG$_{iq}$_2 (18 interactions) than in JSR1-jsG$_{iq}$_1 (31 interactions) (Supplementary Tables 2 and 3). Therefore, we conclude that JSR-jsG$_{iq}$_1 represents a tighter, more engaged conformation of the G protein than JSR-jsG$_{iq}$_2. The plastic and dynamic nature of the receptor-G protein interactions allows slightly different poses of the bound G protein heterotrimer (Supplementary Movie 7).

The quality of both JSR1-jsG$_{iq}$ maps allowed us to model the position of the AHD, which is the most flexible domain in most of the reported GPCR-G protein complexes solved by cryo-EM. As these two maps of JSR1-jsG$_{iq}$ represent the most populated conformations in the purified protein sample, it suggests that the AHD of jsGα$_{iq}$ is instead relatively fixed when coupling to JSR1 (Supplementary Movie 8 and Supplementary Fig. 16). The AHD sits in a semi-open conformation relative to the Ras domain, slightly more open in JSR1-jsG$_{iq}$_2 than in JSR1-jsG$_{iq}$_1 (17° rotation toward the αN helix). In both cases, the αA helix is in proximity to blades 2 and 3 of the Gβ subunit (Supplementary Fig. 16e, f). These interactions might stabilize AHD in this position. Overall, the position of the AHD is similar to that observed in other GPCR-hG$_i$ structures in which the AHD was modeled (Supplementary Fig. 16d).

## Discussion

In monostable bovine rhodopsin, the Meta-II signaling state has a lowered pKa of the Schiff base resulting in deprotonation, which is followed by hydrolysis[41]. On the other hand, JSR1 can maintain a thermostable Schiff base in both the inactive and active states, resulting in bistability[2]. We observe a considerable remodeling of the retinal binding pocket upon activation and, therefore, changes in the hydrogen bond network surrounding the Schiff base, as seen previously in bovine rhodopsin[13,27]. Due to such conformational rearrangements, Glu194$^{45.44}$, Ser199$^{45.49}$, and Tyr126$^{3.28}$ no longer form a continuous hydrogen bond network linking the PSB to the counterion. As Glu194$^{45.44}$ is known to retain its role as counterion in both inactive and active states[12], we propose a model in which water molecules stabilized by Tyr293$^{6.51}$ and Glu194$^{45.44}$ contribute to an extended hydrogen bond network linking the PSB and the counterion in the active state (Fig. 6). The model is supported by resonance Raman experiments showing the presence of a water molecule that hydrogen bonds to the PSB in both the inactive 9-cis and active all-trans states[16]. While the inactive state model is based on the crystal structure of JSR1 bound to 9-cis retinal, Fourier transform infrared spectroscopy analysis suggests that the hydrogen bond network around the retinal is similar when the endogenous chromophore, 11-cis retinal, is bound[42].

We have previously shown that the Glu194$^{45.44}$Asp mutant causes a blue shift of 8 nm[12], supporting this model. In addition, we observed the rotamer change of Ser199$^{45.49}$ away from the water network upon activation. This observation is consistent with mutagenesis data showing that altering the polar nature of Ser199$^{45.49}$ lowers the pKa of the Schiff base in the inactive state[12]. However, while the Ser199$^{45.49}$Phe mutant dramatically shifts the λ$_{max}$ to 380 nm in the inactive state, the active state λ$_{max}$ is unaffected. This is consistent with this position being important only for stabilizing the inactive state PSB (Supplementary Fig. 17). Despite remodeling of the hydrogen bond network upon activation, the Schiff base remains protonated throughout the whole wild-type JSR1 photocycle[12,16] and therefore cannot be hydrolyzed.

Our structures show that the major activation microswitches (such as the C-W-x-P, P-I-F, N-P-x-x-Y, and E/D-R-Y motifs) follow the structural changes previously observed in other class A vertebrate GPCRs (Figs. 3, 4 and Supplementary Figs. 12 and 13). This demonstrates that both invertebrate and vertebrate opsins (or vertebrate class A GPCRs) have similar evolutionary conserved activation mechanisms. Interestingly, the conformation of the E/D-R-Y motif and the surrounding residues in JSR1 is more similar to the human β2-

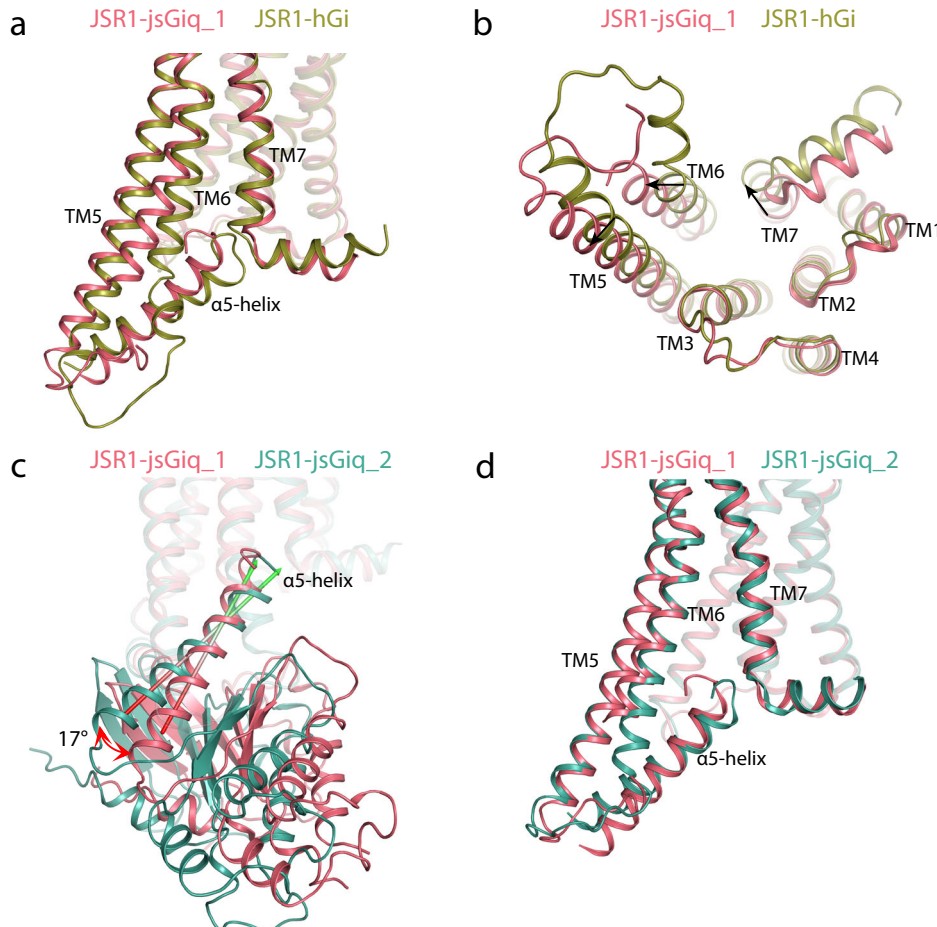

**Fig. 5 | G protein binding site of the JSR1-hG$_i$ complex and the two conformations of the JSR1-jsG$_{iq}$ complex. a** Comparison of the G protein binding site between JSR1-hG$_i$ (olive) and JSR1-jsG$_{iq}$ (salmon). Only the α5 helix of the Gα subunit is shown for the G protein. **b** View from the cytoplasmic side. **c** Relative rotation of the α5 helix between the two conformations of the JSR1-jsG$_{iq}$ complex (JSR1-jsG$_{iq}$_1 in salmon; JSR-jsG$_{iq}$_2 in teal). The main axes of the α5 helices are shown as arrows to depict the relative rotation. **d** Comparison of the G protein binding site between JSR1-jsG$_{iq}$_1 (salmon) and JSR-jsG$_{iq}$_2 (teal). Only the α5 helix of the Gα subunit is shown for the G protein.

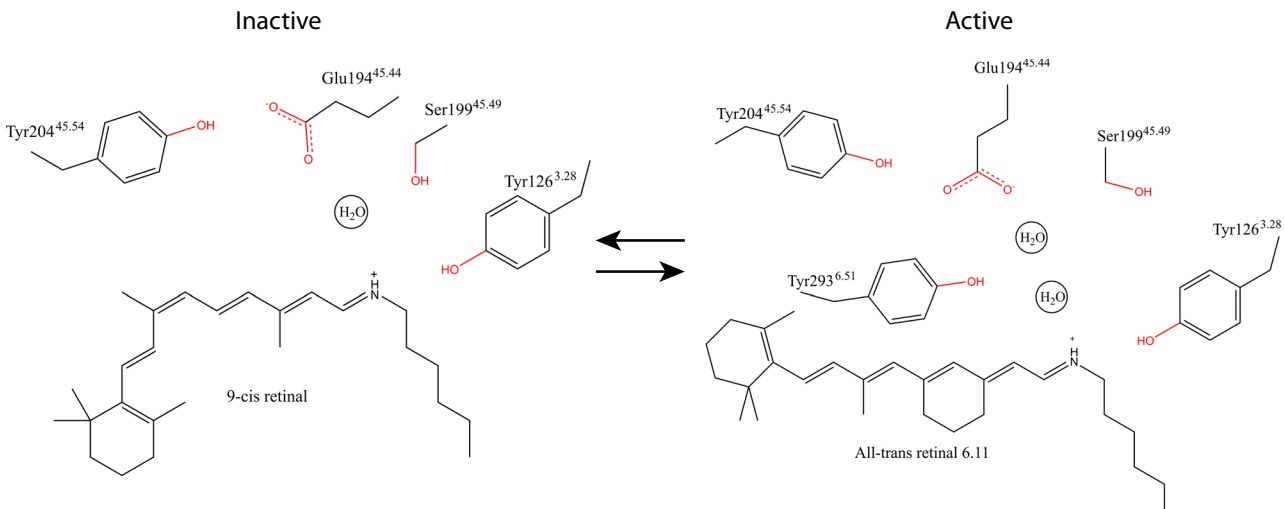

**Fig. 6 | Schematic representation of the PSB – counterion link in the inactive state (left) and active state (right) of JSR1.** In the inactive state, the PSB – counterion link is mediated by a hydrogen bond network between Glu194[45.44], Ser199[45.49], a water molecule, and the PSB[11]. We propose that Ser199[45.49] does not participate in this network in the active state. Instead, the PSB – counterion link is mediated by a similar water-mediated hydrogen bond network between Glu194[45.44], two water molecules, the PSB, and possibly Tyr293[6.51].

adrenergic receptor than to bovine rhodopsin (Supplementary Fig. 13). This indicates that, in some respects, bistable opsins are structurally more similar to non-photosensitive human class A GPCRs than to monostable opsins. Interestingly, our structures suggest that the extent of the rearrangement of TM6 is modulated by the packing of TM6 and TM7 near the C-W-x-P and N-P-x-x-Y motifs. In $G_i$-coupled conformations, tighter packing seems to restrict the extent of the TM6 movement. We hypothesize that selectivity for $G_i$ proteins could be achieved through a stable packing in TM6/TM7 that locks TM6, preventing it from moving further, as observed in bovine rhodopsin (Supplementary Fig. 18, **top right**). In $jsG_{iq}$-coupled conformations, a looser packing in this region allows for a larger movement of TM6 that can accommodate slightly different poses of the C-hook in a wider and more plastic receptor binding pocket.

Our analysis of the JSR1-$hG_i$ and JSR1-$jsG_{iq}$ complexes reveals several differences in the binding pose of the G proteins. Compared to other class A GPCR-$hG_i$ complexes, the JSR1-$hG_i$ complex shows a somewhat unusual binding pose of the human $G\alpha_i$. The $\alpha5$ helix adopts a somewhat shallower binding pose compared to other GPCR-$hG_i$ complexes, and JSR1 interacts differently with the $hG_i$ than has been observed in other GPCR-$hG_i$ structures. Nevertheless, our in vitro activity assays clearly show activation of the $hG_i$ by JSR1 (Fig. 1b). The shallow binding pose likely arises from the differences in the amino acid sequences of the $\alpha5$ helices of $hG_i$ and the visual jumping spider $G_{q1}$ (Supplementary Fig. 5a). JSR1 is able to couple to $hG_i$ and activate it, but it might do so through slightly different interactions than vertebrate GPCRs. On the other hand, our JSR1-$jsG_{iq}$_1 complex shows a fully engaged $G\alpha_{iq}$, as most of the reported GPCR-G protein complexes. The JSR1-$jsG_{iq}$_2 complex structure shows a G protein rotation by 17° and an AHD that has moved further away from the Ras domain compared to those in JSR1-$JSG_{iq}$_1 (Fig. 5c and Supplementary Movies 2 and 7). The presence of two different conformations shows the plasticity of the G protein binding as well as the conformational flexibility of the G protein. The nucleotide exchange in GPCR-G protein complexes is a dynamic process, where the AHD is highly flexible while the Ras domain is restrained by the receptor. These two captured snapshots represent two of the conformations of this dynamic process even in a nucleotide-free condition[43]. Our work shows that despite the slight differences in the structures of the JSR1-hGi and JSR1-$jsG_{iq}$ complexes, the receptor is capable of activating the G protein and inducing nucleotide exchange in both G proteins.

A particularly interesting region in our structures is ICL3. Unlike in many existing GPCR-G protein complexes, we observe a relatively ordered ICL3, including the cytoplasmic ends of TM5 and TM6. Still, in three of the four existing structures of JSR1 there are missing or poorly resolved densities at different positions in this region (Supplementary Fig. 15a). Most of ICL3 is missing in the inactive JSR1 structure[11], but in the JSR1-$jsG_{iq}$ and JSR1-$hG_i$ complexes, ICL3 shows different degrees of interaction with $G\alpha$ at the region of the $\alpha4$ helix and $\beta6$ strand (Supplementary Fig. 15b), despite $hG\alpha_i$ and $jsG\alpha_{iq}$ have the same sequence in this region. This suggests that ICL3 is flexible in order to accommodate diverse binding poses of $G\alpha$, and it may, therefore, contribute to regulating the enzymatic activity of $G\alpha$. In our JSR1-$hG_i$ structure, we see a similar outward movement of TM5 and TM6 compared to the monostable bovine rhodopsin-$G_i$/$G_t$ structures[24,25,44], and in the JSR1-$jsG_{iq}$ structures, the outward movement of TM5 and TM6 is even larger (Supplementary Fig. 14). Our structures indicate that the G protein subtype determines the extent of the TM6 movement rather than the mono- or bistability properties of the receptor. For optogenetic applications, further experiments will be needed to determine the signaling profile of JSR1 with different human G protein subtypes and homologs, e.g., to measure G protein activation efficiency profiles.

Bistable opsins are promising for optogenetic applications because they can be activated and inactivated with specific wavelengths of light, providing high temporal precision. For instance, JSR1

has been used as an optogenetic tool in zebrafish reticulospinal V2a neurons, where activation of JSR1 leads to an increase of $Ca^{2+}$ and evokes a change in swimming behavior[45]. The main issue with using JSR1 as an optogenetic tool is that the $\lambda_{max}$ of the active and inactive states overlap, meaning that there will be a mixture of active and inactive states upon illumination, and the activity of JSR1 cannot be controlled precisely[16]. Interestingly, the JSR1 $S199^{45.49}F$ mutant has a $\lambda_{max}$ of 540 nm in the active state (similar to WT) but 380 nm in the inactive state, due to the deprotonation of the Schiff base[12]. $Ser199^{45.49}$ is involved in the Schiff base-counterion link in the inactive state but not in the active state, as we show in this work, which explains the significantly blue-shifted $\lambda_{max}$ in the inactive state and the unaffected $\lambda_{max}$ in the active state. The separation of the $\lambda_{max}$ between inactive and active states of JSR1 by the $S199^{45.49}F$ mutation enables precise optical control of the receptor. Interestingly, this JSR1 $S199^{45.49}F$ mutant has been demonstrated as an effective optogenetic tool to modulate neuronal activity in zebrafish[45]. However, UV-light activation is not very desirable for in vivo applications because it does not penetrate tissues well and can induce DNA damage[46]. Thus, further engineering of JSR1 is still required to optimize its spectroscopic properties as an optogenetic tool.

In this work, we disclose the first structures of activated signaling complexes between a bistable opsin and G proteins. We foresee that our structures will facilitate the rational engineering of bistable opsins to manipulate their spectroscopic properties and signaling profiles and to develop optimized optogenetic tool proteins.

## Methods

### Synthesis of retinal analogs
All-trans retinal 6.11 was synthesized as described previously[47].

### JSR1 Expression and purification
Wild-type Jumping Spider Rhodopsin isoform-1 (JSR) from *Hasarius adansoni* tagged with a C-terminal 1D4 epitope[15], was recombinantly expressed in HEK293 GnTI⁻ cells as described previously[16]. Cells were harvested by centrifugation at $500 \times g$, and the pellets were stored at $-80\,°C$. The cells were later thawed in lysis buffer (50 mM HEPES pH 6.5, 150 mM sodium chloride, 3 mM magnesium chloride) supplemented with 1 cOmplete protease inhibitor cocktail tablet per 50 ml of lysis buffer (Roche) and mechanically disrupted in a Dounce homogenizer. All the following steps were performed under dim red light conditions. Retinal was added to the lysate (6.11 all-trans-retinal (see section above) or 9-cis retinal (Sigma-Aldrich) to a final concentration of 10 μM and the suspension was incubated at 4 °C overnight. Dodecyl-β-D-maltoside (DDM) powder (DDM Solgrade; Anatrace) was then added to a final concentration of 1% (w/v) to solubilize cell membranes for 2 h at 4 °C and the unsolubilized fraction was removed by centrifugation at $100,000 \times g$. The supernatant was incubated with Sepharose resin (GE Healthcare Life Science) coupled to 1D4 antibody[44](Cell Essentials) at 4 °C overnight, where the resin contains 3.7 mg 1D4 antibody per ml resin. The resin was then loaded into a glass Econo-Column (Bio-Rad Laboratories, CA), and washed with 30 column volumes of wash buffer (50 mM HEPES pH 6.5, 150 mM sodium chloride, 3 mM magnesium chloride, 0.01% DDM (w/v)) before the addition of 1.5 column volumes of elution buffer (50 mM HEPES pH 6.5, 150 mM sodium chloride, 3 mM magnesium chloride, 0.01% DDM (w/v), 800 μM 1D4 peptide TETSQVAPA (Peptide 2.0) to the resin. The slurry was incubated at 4 °C overnight, and the protein was eluted from the column the following day. Residual protein was eluted from the resin after the addition of three more column volumes of wash buffer. All eluted fractions were pooled and concentrated using a 50 kDa molecular weight cut-off Amicon Ultra Centrifugal Filter (Millipore). To the JSR1 protein sample for forming the JSR1-$hG_i$ complex, an additional wash step was applied to exchange detergent from 0.01% DDM to 0.01% lauryl maltose neopentyl glycol (LMNG).

## Gα$_{i1}$β$_1$γ$_1$ Expression and purification

Human Gα$_i$ subunit (Gα$_{i1}$) with an N-terminal TEV protease-cleavable deca-histidine tag was expressed in *E. coli* BL21 (DE3) cells (Sigma-Aldrich) and purified as described previously[48]. The Gβ$_1$γ$_1$ subunits were separated from the transducin G protein heterotrimer, which was purified from bovine retinae as described by Maeda et al.[49]. Human Gα$_{i1}$ and bovine Gβ$_1$γ$_1$ were mixed to generate the Gα$_{i1}$β$_1$γ$_1$ heterotrimer used for complex formation with JSR.

## Protein expression and purification

**Gα$_{iq}$β$_1$γ$_2$ expression.** Mutations were introduced to the human Gα$_i$ to match the sequence of the jumping spider Gα$_{q1}$ (HaG$_{q1}$: acc. No. LC799818) (A31R; D193S; L194I; [residues 337–354 = DAVTD-VIIKNNLKDCGLF] Gα$_{i1}$ to [residues 337–354 = CAVKDTILQNNLK-ECNLV]) (Supplementary Fig. 5). The Gα$_{iq}$ chimera, hGβ$_1$ and hGγ$_2$ subunits were cloned into the pAC8RED vector for insect cell expression[50].

Baculoviruses were produced in Sf9 insect cells using the FlashBac technology (Oxford expression technologies). $1.0 \times 10^6$ Sf9 cells were co-transfected with 1.5 µg plasmid DNA and 500 ng linearized Bac10:KO1629 viral DNA using Cellfectin II (Invitrogen) following the manufacturer's protocol. Cells were incubated for five hours at 27 °C with the transfection mixture before it was replaced by fresh SF900II SFM insect cell medium (Invitrogen) supplemented with 1% Pen/Strep (PAN Biotech). After seven days of incubation at 27 °C, the V$_0$ generation virus was harvested by centrifugation at $800 \times g$ and 20 °C, the supernatant supplemented with 1% Pen/Strep and 10% FCS (Sigma-Aldrich) and stored in the dark at 4 °C.

Viruses were two times amplified by infecting new Sf9 cultures at $2.0 \times 10^6$ cells/ml in SF900II SFM medium with 1% V$_0$ or V$_1$ generation viruses, respectively. V$_1$ or high-titer V$_2$ generation viruses were harvested 72 h post-infection by centrifugation as described above. However, only 1% of FCS was used as a supplement.

The jsG$_{iq}$ heterotrimer was expressed in High Five insect cells (Invitrogen). Large-scale expression was conducted in High Five insect cells at a density of $4.0 \times 10^6$ cells/ml in SF900II SFM medium using 5 L Erlenmeyer flasks (Corning). Prior to expression, the medium was exchanged, and cells were resuspended at $4 \times 10^6$ cells/ml in a fresh SF900II SFM medium. Then, Baculovirus was added in an optimal ratio and cells were cultured at 27 °C and 120 rpm and harvested 48 h post-infection by centrifugation. Pellets were flash-frozen in liquid nitrogen and stored at − 80 °C.

**Gα$_{iq}$β$_1$γ$_2$ purification.** The frozen cell pellet was thawed and resuspended in a buffer containing 20 mM HEPES pH 7.5, 100 mM NaCl, 3 mM MgCl$_2$, 100 µM EDTA, 5 mM β-mercaptoethanol and 20 µM GDP supplemented with DNase I (10 µg/ml) and cOmplete protease inhibitor cocktail tablets (Roche). Cells were mechanically disrupted using a Microfluidizer (Microfluidics International Corporation) and subsequently centrifuged at 40,000 rpm ($185,000 \times g$) for 45 min. The pellet was resuspended in 20 mM HEPES, 100 mM NaCl, 20 mM imidazole, 3 mM MgCl$_2$, 5 mM β-mercaptoethanol and 20 µM GDP supplemented with cOmplete protease inhibitor cocktail tablets and solubilized by adding 1% Sodium cholate (Sigma-Aldrich) and 0.05% (w/v) DDM (Anatrace). After incubation at 4 °C for 1 h, insoluble material was removed by centrifugation at $100,000 \times g$ for 45 min. The clarified supernatant was incubated with 5 ml of TALON resin (Takara Bio) for 90 min at 4 °C. The resin was collected in a gravity flow column, and the detergent was gradually exchanged for 0.1% DDM. The resin was washed with 10 column volumes of IMAC buffer A (20 mM HEPES pH 7.5, 100 mM NaCl, 20 mM imidazole, 1 mM MgCl$_2$, 5 mM β-mercaptoethanol, 20 µM GDP, and 0.01% DDM) and 10 column volumes of IMAC buffer B (IMAC buffer A with 300 mM NaCl). The sample was eluted with IMAC buffer A supplemented with 300 mM imidazole. His-tagged HRV 3 C protease was added (1:50 w/w; HRV 3 C protease: jsG$_{iq}$

heterotrimer) to the eluent and dialyzed against dialysis buffer (20 mM HEPES pH 7.5, 100 mM NaCl, 1 mM MgCl$_2$, 100 µM TCEP, 20 µM GDP and 0.05% DDM) overnight at 4 °C. Dialysate was collected and supplemented with imidazole to a final concentration of 20 mM before incubation with 5 ml TALON resin for 1 h at 4 °C. The resin was transferred in a gravity flow column and the flow through was collected. The resin was washed with one CV of dialysis buffer supplemented with 20 mM imidazole. The flow through and wash were combined and loaded to a HiTrap Q FF 1 ml column (Cytiva). The column was washed with 15 CVs of Q buffer A (20 mM HEPES pH 7.5, 100 mM NaCl, 1 mM MgCl$_2$, 100 µM TCEP, 20 µM GDP, and 0.05% DDM), and the sample was eluted with a linear gradient over 30 CVs with Q buffer B (Q buffer A with 1000 mM NaCl). The relevant fractions were pooled and concentrated using a 50 kDa cut-off Amicon Ultra Centrifugal Filter (Millipore). The concentrated sample was injected into a SRT-C 300 SEC column (Sepax Technologies) pre-equilibrated with SEC buffer (20 mM HEPES pH 7.5, 100 mM NaCl, 1 mM MgCl$_2$, 100 µM TCEP, 20 µM GDP and 0.02% DDM). Fractions containing the jsG$_{iq}$ heterotrimer were pooled and concentrated using a 50 kDa cut-off Amicon Ultra Centrifugal Filter (Millipore) before adding 10% glycerol (v/v). The sample was flash-frozen in liquid nitrogen and stored at − 80 °C.

## Complex formation and size exclusion chromatography

The 1D4 purified receptor was mixed with purified jsG$_{iq}$ or hG$_{i1}$ heterotrimer at a molar ratio of 1:1.25 and supplemented with apyrase (New England Biolabs) (25 mU/mL) to degrade any GTP and GDP present in the solution. After one hour of incubation at 4 °C, the JSR1-jsG$_{iq}$ sample was concentrated using a 100 kDa cut-off Amicon Ultra Centrifugal Filter (Millipore) and injected into an SRT-C SEC 300 (10/300) column (Sepax Technologies) pre-equilibrated with SEC buffer (20 mM HEPES pH 6.5, 100 mM NaCl, 0.01% DDM and 1 mM MgCl$_2$). The JSR1-jsG$_{iq}$ sample was only exposed to ambient room light during injection to the SEC purifier, and during fraction collection, the sample was protected from light while on the column due to the opaque nature of the HPLC column casing. The JSR1/hG$_i$ mixture was illuminated with 495 nm long-pass filtered light under light-controlled conditions to induce 9-cis to all-trans-retinal isomerization, followed by 30 min incubation in the dark. The JSR1-hG$_i$ sample was concentrated and subjected to a Superdex 200 Increase (10/300) column equilibrated in 20 mM HEPES pH 7.5 and 100 mM NaCl. Relevant fractions were pooled and concentrated to 1 mg/ml (JSR1-hG$_i$) and 8 mg/ml (JSR1-jsG$_{iq}$).

## Cryo-EM grid preparation

A volume of 3 µl purified protein complex at a concentration of 1 mg/ml (JSR1-hG$_i$) or 8 mg/ml (JSR1-jsG$_{iq}$) was applied to glow-discharged holey carbon grids (Quantifoil R1.2/1.3 Cu 200 mesh). The grids were blotted with a Vitrobot Mark IV (Thermo Fisher Scientific) and flash-frozen in liquid ethane. The sample was blotted for 3 s (JSR1-hG$_i$) or 6 seconds (JSR1-jsG$_{iq}$) at 4 °C and 100% humidity. The grids for the JSR1-hG$_i$ complex were blotted and frozen under dim red light conditions to limit light exposure.

## Cryo-EM data acquisition

Data was acquired on a 300 kV Titan Krios (Thermo Fisher Scientific) equipped with a GIF Quantum LS energy filter (Gatan), operated with an energy slit width of 20 eV. Movies were recorded with a K3 direct electron detector (Gatan) with a dose rate of 19 e$^-$/px/s (40 frames/micrograph), and target defocus values from − 1.0 to − 2.4 µM in 0.2 µM intervals in an automatic way using EPU software (Thermo Fisher Scientific). For the acquisition of the JSR1-jsG$_{iq}$ complex, a pixel size of 0.51 Å/pixel and a total dose of 70 e$^-$/Å$^2$ was used, and for the acquisition of the JSR1-hG$_i$ complex, a pixel size of 0.85 Å/pixel and a total dose of 50 e-/Å$^2$ was used.

## Single-particle analysis of the JSR-jsG$_{iq}$ complex

Cryo-EM data was processed using cryoSPARC (Structura Biotechnology Inc)[51] and Relion4[52]. Pre-processing was done on the fly using MotionCor2[53] with dose-weighting, and micrographs were binned by two, resulting in a pixel size of 1.02 Å. Motion-corrected micrographs were then imported to cryoSPARC and contrast transfer function (CTF) parameters were determined by patch CTF estimation (multi). Micrographs with a CTF fit resolution > 8 Å were discarded. Initial particles were picked on a subset of all micrographs using Topaz[54] with the general model ResNet8, and the particles were subjected to 2D classification. 2D class averages showing the JSR-jsGiq complex with the corresponding particles were selected to train a model for automated particle picking in Topaz. Automated picking on the full dataset yielded 2,964,739 particles. Particles were again subjected to 2D classification. A total of 768,100 particles grouped into 2D class averages showing the JSR-jsG$_{iq}$ heterotrimer complex. Ab initio models were generated with cryoSPARC and 3D classification was performed to separate different conformations of the complex. Particles from the two best classes were used for non-uniform refinement and yielded two maps with 4.06 Å (159,665 particles) and 4.17 Å (134,167 particles) resolution. Although local CTF refinement and subsequent non-uniform refinement yielded maps with similar resolution according to the Fourier Shell Correlation (FSC) at FSC = 0.143, visually, the quality of the maps improved. CryoSPARC's 3D variability analysis[55] was used to visualize motion in the particle set.

## Single-particle analysis of the JSR-hGi complex

The JSR1-hG$_i$ complex was processed slightly differently: movies were binned, dose-weighted, and corrected for beam-induced motion with the Relion implementation of MotionCor2[53], and CTF estimation was done with CTFFIND4.1[56]. Final particle picking was performed in crYOLO 1.6[57] using a general model. A total number of 8.4 million particles were extracted with binning by three. Particles were subjected to 2D classification in Relion to remove false positive and bad quality picks. A set of 1.2 million good-quality particles (14% of initially picked particles) was subjected to 3D classification with a soft mask. Two classes showed clear secondary structure features for the 7TM region. Particles corresponding to both classes were re-extracted at original sampling and subjected to 3D refinement coupled with SIDESPLITTER[58], with a soft mask excluding the AH domain, followed by particle polishing and final 3D refinement coupled with SIDESPLITTER. The final 3D reconstruction was subjected to post-processing with a soft mask excluding the Gα AH domain, leading to an overall 4.9 Å resolution.

## Model building and refinement JSR-hG$_i$

A sharpened and locally filtered volume was rigidly fitted in UCSF Chimera[59] with dark state JSR1 (6I9K), Gα$_{i1}$ (PDB: 6PT0), and Gβ$_1$γ$_1$ dimer (PDB: 6OY9). MDFLEX was used initially with default settings for a few iterations to adapt the model to the conformational changes in JSR1.

## Model building and refinement JSR-jsG$_{iq}$

The initial model of JSR was obtained from the crystal structure of the inactive state (6I9K), and the initial model for the jsG$_{iq}$ heterotrimer was obtained from the crystal structure of the human G$_i$ heterotrimer (6CRK). Residues in the Gαi subunit were mutated to match the jsG$_{iq}$ sequence. The models were docked into the electron density maps as rigid bodies using UCSF Chimera[59]. Realspace refinement with phenix.real_space_refine in the Phenix suite[60] and interactive model building in Coot[61] was performed iteratively. The H8 was manually built in Coot as it was not modeled in the crystal structure (6I9K).

## GTPase Glo assay® (Promega)

Reactions were set up in 384-well, low-volume, PS, white, solid-bottom plates (Greiner Bio-One, Kremsmünster, Austria) and carried out at 20 °C under dim-red light conditions. Each condition was prepared in octuplicates. 2.5 μl of protein solution was mixed with 2.5 μl of 2 μM GTP solution. Reference samples contained either 2 μM G protein or 0.2 μM JSR1 with either 9-cis retinal or 6.11 all-trans retinal analog (dark and light-activated). Test samples contained 2 μM G protein and 0.2 μM JSR with the respective ligand (dark and light-activated). A control sample contained no proteins. The plate was sealed and incubated for 30, 60, or 120 minutes shaking at 500 rpm. Subsequently, 5 μl of GTP-Glo reagent was added to each well, and the reaction was carried out for 30 min. Finally, 10 μl of Detection Reagent was added to each well. Luminescence was measured after 10 min in a PheraStar FSX (BMG Labtech, Ortenberg, Germany) plate reader (Optical module: LUM plus, Gain: 3600, focal height: 14.5, 1 second/well). Outliers were identified using the Grubbs-test[62]. GTP hydrolysis was calculated in relation to the control sample:

$$GTP\ hydrolysis\,[\%] = \left( \sum_{1}^{n} 100 - \frac{Lum(x)_n}{Mean\ Lum_0} \times 100 \right) / n$$

where $Lum_0$ is the luminescence signal of the control sample and $Lum(x)$ the luminescence of each condition $(x)$ and $n$ the number of replicates.

## Sequence alignment and conservation

For the sequence alignment and conservation analysis of vertebrate opsins, the human rhodopsin sequence was used as a query for a BLASTp search using reference proteins, including only vertebrate sequences. In total, 202 vertebrate opsin sequences were selected and aligned using Clustal Omega[63]. The output file was then used as input to WebLogo[64]. For the sequence alignment and conservation analysis of invertebrate opsins, the jumping spider rhodopsin sequence was used as a query for a BLASTp search using reference proteins excluding vertebrate sequences. In total, 113 invertebrate opsin sequences were selected. Alignment and creation of the WebLogo plot were done the same way as for the vertebrate opsin sequences.

## Structure alignment

Structure alignments and comparisons were done in PyMOL[65]. Receptor structures were aligned with conserved residues inside the transmembrane region of the receptors. The following residues were used for alignment (Ballesteros-Weinstein numbering[66]): 1.45–1.54, 2.46–2.56, 3.34–3.44, 4.48–4.56, 7.38–7.46.

Protein Data Bank accession codes for the structures used for comparison are listed below:

Inactive GPCRs: JSR1 (6I9K), bovine Rhodopsin (1GZM), squid rhodopsin (2Z73).

Active GPCR-G protein complexes: bovine rhodopsin-Gα$_t$ peptide (4A4M), bovine rhodopsin-Gα$_t$ peptide (3PQR), bovine rhodopsin-G$_i$ (6CMO), bovine rhodopsin-mini G$_o$ (6FUF), bovine rhodopsin-Gα$_t$ peptide (5EN0), β2-adrenergic receptor-G$_s$ (3SN6), M1-receptor-G$_{11}$ (6OIJ), NTSR1-G$_i$ NC (6OSA).

For the AHD comparison, the Ras domain of the Gα subunits (Residues 31–58 and 181–354) was aligned. Accession codes for the structures used for comparison are listed below:

GABA (B) receptor-human G$_i$ complex (7EB2), NTSR1-human G$_i$ complex (7L0S), cannabinoid receptor 2-human G$_i$ complex (6PT0).

## Protein sequences

Exact protein sequences and their sources are summarized in Supplementary Table 4.

## Reporting summary

Further information on research design is available in the Nature Portfolio Reporting Summary linked to this article.

## Data availability

Atomic coordinates of JSR1-jsG$_{iq}$_1, JSR1-jsG$_{iq}$_2, and JSR1-hG$_i$ without the AHD have been deposited to the Protein Data Bank under accession codes 9EPP, 9EPQ and 9EPR, respectively. Cryo-EM maps of JSR1-jsG$_{iq}$_1, JSR1-jsG$_{iq}$_2, and JSR1-hG$_i$ have been deposited in the Electron Microscopy Data Bank under accession codes EMD-19882, EMD-19883 and EMD-19884, respectively. Source data are provided as a Source Data file. Source data are provided in this paper.

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

## Acknowledgements

We would like to thank Dr. Elena Lesca, Dr. Niranjan Varma, and Mara Wieser for their contribution to JSR1 biochemistry. We would like to thank Jonas Mühle for his technical support for the purifications and the GTPase Glo assay. We would like to thank Dr. Jacopo Marino for his support with cryo-EM data collection and analysis, as well as Dr Francisco Leisico and Lea Jänisch for their input in the manuscript. We would like to thank the staff at the Scientific Center for Optical and Electron Microscopy at ETH Zurich for access and excellent user support. We would like to thank Dr. Daniel Boehringer of the Cryo-EM Knowledge Hub at ETH Zurich for his support of cryo-EM experiments. M.S. thanks the Kimmelman Center for Biomolecular Structure and Assembly for partial support. M.S. holds the Katzir-Makineni Chair in Chemistry. This project has received funding from the European Research Council (ERC) under the European Union's Horizon 2020 research and innovation program (Grant agreement No. 951644 to G.F.X.S., and Marie Skłodowska-Curie grant agreement No. 701647 to M.J.R.); the project was also funded by Swiss National Science Foundation grants (No. 192760 and No. 183563 to G.F.X.S.), by the Synapsis Foundation Switzerland (2018-PI04 to G.F.X.S.), and by the JSPS KAKENHI (No. 23H02516 and 21H04969 to A.T., and No. 22H02663 to M.K.).

## Author contributions

G.F.X.S., M.J.R., M.S. and C.-J.T. conceived the project. T.N., M.K. and A.T. sequenced the jumping spider $G_q$ genes. O.T., F.P., M.J.R. and C.-J.T. purified the proteins. I.D. synthesized the all-trans retinal 6.11. O.T. performed the GTPase Glo assay. O.T., F.P. and C.-J.T. prepared the samples for cryo-EM. O.T., F.P., and P.A. collected data at the microscope. O.T., F.P. and P.A. performed single-particle analysis. O.T., F.P., M.J.R., and C.-J.T. built the atomic models. O.T., F.P., X.D., G.F.X.S., M.J.R. and C.-J.T. analyzed the experimental data. O.T., X.D., G.F.X.S., M.J.R. and C.-J.T. wrote the manuscript. All authors reviewed and contributed to the manuscript.

## Competing interests

G.F.X.S. declares that he is a co-founder and scientific advisor of the company leadXpro AG and InterAx Biotech AG. All other authors declare no competing interests.
