## [Transparent Peer Review file · Nature Communications]

Active state structures of a bistable visual opsin bound to G proteins

Corresponding Author: Dr Ching-Ju Tsai

Version 0:

Reviewer comments:

Reviewer #1

(Remarks to the Author)

The manuscript by Tejero et al. reports the cryo-EM structures of the jumping spider rhodopsin isoform-1 (JSR1) in complex with the human Gi and chimeric jsGiq heterotrimers. These structures provide insights into different activation mechanisms for the monostable and bistable opsins. However, one major concern is that the authors do not provide enough complementary data, such as receptor mutagenesis, binding and signaling data, to support their findings. Furthermore, a few points should be addressed:

1. The authors used a human Gi/jumping spider Gq chimera (jsGiq), which was generated by engineering the hGi to match the AA sequence of the jumping spider visual Gq1 in the receptor binding interface. However, this chimeric G protein is not biological relevant. Have the authors tried the jumping spider Gq instead of the chimera?
2. For the JSR1-hGi and JSR1-jsGiq cryo-EM data processing, the authors used different pipeline softwares, cryoSPARC for JSR1-jsGiq and Relion for JSR1-hGi. Have the authors tried to use the same software for data processing? As to my concern, Relion and cryoSPARC may give different results in some cases. So, it is worthwhile to check if one or the other software may give better result. Meanwhile, it was noticed that the particles in the JSR1-hGi dataset were polished in the final refinement, while not in the JSR1-jsGiq dataset. Why is that? If the authors did not polish the particles in the JSR1-jsGiq dataset, this should be done, as this process usually increases map resolution and quality.
3. In the JSR1-jsGiq dataset, more than 74k images were collected. And a total number of 8.4 million particles were picked and extracted for later processing. The authors obtained two final maps with 134k and 159k particles. It is not unexpected to have many particles discarded, but for full clarity, the authors should modify the workflow in Supplementary Figure 5 by displaying more information and intermediate maps. Also, angular distribution plot should also be included. The same should be applied to the JSR1-hGi dataset in Supplementary Figure 3.

Minor:

1. Lines 230-242, Glu19445.54 should be Glu19445.44, as Glu19445.44 is used for the numbering scheme in the rest of the paper, including P15, Fig. 2, Fig. 6, Supplementary Fig. 6-7.
2. Throughout the text, a subscript should be used for G protein subtype expression, e.g. G_q instead of Gq.

Reviewer #3

(Remarks to the Author)

The manuscript entitled "Active state structures of a bistable visual opsin bound to G proteins" by Tejero et al. gives the first insight into the structure of the active state of bistable rhodopsin and its formation and suggests a potential G protein selectivity mechanism.

Animal rhodopsins are G protein-coupled receptors and photo-sensitive pigments that contain an 11-cis retinal chromophore. Upon light absorption, isomerization of 11-cis retinal to its all-trans form induces protein conformational changes that activate the GPCR and allow signal transmission to the G protein. Light-activated vertebrate rhodopsin loses the all-trans-retinal due to hydrolysis of the Schiff base linkage and requires binding of 11-cis retinal to restore rhodopsin. In contrast, invertebrate rhodopsins do not lose the retinal. They rather absorb a second photon to isomerize the retinal back to its initial 11-cis state and are therefore called bistable rhodopsins, i.e. stable in the inactive and active state. So far it was not possible to obtain a structure of invertebrate rhodopsin in the active state, because of the largely overlapping absorption

profiles of the inactive and active states that typically cause a mixture of these states. The authors Tejero et al. used two approaches to obtain complexes between jumping spider rhodopsin 1 (JSR1) and G protein. Reconstituting JSR1 with 9-cis retinal caused a 30 nm spectral blue shift, which allowed specific illumination to enrich all-trans-retinal containing active JSR1 that could be complexed with the human inhibitory G protein (hGi) for structure analysis by cryo-electron microscopy (cryo-EM). In a second approach JSR1 was reconstituted from the opsin apo protein and a retinal analog, i.e. all-trans-retinal with a 6-carbon ring locking the C11-C12 double bond in the trans configuration (ATR6.11). In contrast to all-trans-retinal, ATR6.11 did form a stable pigment with opsin, which is capable to activate G protein and form a complex with a human Gi/jumping spider Gq chimera (jsGiq) for cryo-EM analysis.

The authors present a cryo-EM map of JSR1-hGi complex (4.9 Angstrom resolution) and two maps of JSR1-jsGiq complexes (4.06 and 4.17 Angstrom resolution), giving first insight into commonalities and differences of the activation mechanisms of monostable and bistable rhodopsins. The discovery of ATR6.11 as an agonist suitable for JSR1-G protein complex formation is an important step towards understanding the active structure of JSR1 and has big potential to also be useful for other bistable rhodopsins. Bistable rhodopsins are not only employed for vision in invertebrates, but also serve in non-visual light perception in vertebrates, and therefore the elucidation of their activation mechanism is of great interest in this field.

The present work, however, has a couple of points which need clarification and/or additional experiments:

1) The retinal analog ATR6.11 is a key element for the present work. Its biochemical and biophysical characterization as chromophore for JSR1 has been reported in a BioRxiv preprint (Ref. 18). It is important that this characterization is either published separately or as part of the present manuscript.

2) The authors state that the JSR1(ATR6.11)-jsGiq complex sample for cryo-EM study was purified under ambient light conditions. According to the method section, the final SEC purification was done with a SRT-C 300 column for JSR1(ATR6.11)-jsGiq (Supplementary Fig. 1C) and a Superdex 200 increase column for JSR1-hGi (Supplementary Fig. 1A). The SEC profile of JSR1(ATR6.11)-jsGiq doesn't show well separation of the complex. Peaks 1 and 2 are barely separated and contain the same proteins. Is this an effect of the different columns used? Or is it due to purification under ambient light? Does the non-single peak reflect the two structures observed in cryo-EM structure determination? Where is the unbound G protein peak? Why was not a Superdex 200 column used? These questions arise because Figure 2 of the BioRxiv paper shows good separation for both, the JSR1(ATR6,11)-hGi complex and the JSR1(ATR6.11)-human Gq complex.

It is very confusing that the sample preparation has not been performed in the dark. Even if the retinal is chemically locked at the C11-C12 double bond, there is still the possibility that ambient light may cause trans-cis isomerization of other bonds of ATR6.11 in JSR1. Active cis retinal forms have been described for locked retinal reconstituted bovine rhodopsin and have been discussed by the authors in the BioRxiv preprint (Ref. 9, Gulati et al. 2017 PNAS). The complexes of JSR1(ATR6.11) with hGi and hGq, respectively, in the BioRxiv preprint have been purified in the dark and are well separated by SEC. Good separation is also seen for the JSR1-hGi complex in the present study, which was obtained by illumination of JSR1(9-cis retinal) followed by complex formation with hGi and purification in the dark. Perhaps the light conditions and Superdex column cause the difference for JSR1(ATR6.11) and should be clarified.

3) The BioRxiv preprint shows that ATR6.11 acts as an agonist for JSR1. The activity of JSR1(ATR6.11) is increased when the pigment is illuminated. Given that cis isomerizations of 11-cis locked retinal in bovine rhodopsin cause activity, it would be interesting to know the retinal composition upon illumination of JSR1(ATR6.11). A mixture of isomers could be an explanation for the result of the JSR1(ATR6.11)-jsGiq complex purified under ambient light.

4) The lower resolution of JSR1-hGi complex was explained by a possible mixture of retinal isomers and the reason to use ATR6.11 in the JSR1(ATR6.11)-jsGiq complex. The authors also suggested other factors such as heterogeneity of the complex and the tendency of dissociation for the JSR1-hGi complex. A mixture of all-trans-retinal and 11-cis retinal should yield a mixture of complex and dissociated complex, which particle picking in cryo-EM analysis should be able to separate. It might be possible that the higher affinity of jsGiq might allow stable complex formation. Did the authors test this?

If there is heterogeneity, it may arise from other cis isomers different from 11-cis. Supplementary Figure 6CD shows differences in the density maps for ATR6.11 and all-trans-retinal when present in JSR1. Especially striking is the difference around the C13-C14 bond. The retinal in JSR1-hGi (blue mesh) may contain a mixture of retinals including one with a 13-cis bond, perhaps as a result of the illumination conditions used for 9-cis retinal reconstituted JSR1. Unfortunately, the figure legends lack in several cases clarity about the sample, which needs to be corrected. If 13-cis is present, it would be similar to a potential trans-cis isomerization of ATR6.11. Such a 13-cis isomer could be important as it may have a functional role and requires further investigation.

On page 7, the authors mention that cryo-EM densities of JSR1-hGi and JSR1-jsGiq are similar in pose but there is a slight difference in lysine side chain and the Schiff base. For readers who are interested in residue changes upon activation, however, the different positions of Schiff base wouldn't be trivial. A discussion of the different positions of standard retinal and ATR6.11 Schiff base should be added for clarification. A 13-cis isomer might potentially be an explanation.

Reviewer comments:

Reviewer #1

(Remarks to the Author)

The authors have addressed my concerns.

Reviewer #3

(Remarks to the Author)

The authors adequately responded to my questions/issues raised in my first round of review. The revised manuscript has thoroughly addressed the issues, and their preprint addressing the key element of this manuscript seems to have been accepted already.

The revised manuscript is now easier to follow based on feedback from the reviewers. I would advise that it now be accepted for publication.

REPLIES TO THE REVIEWER'S COMMENTS

We thank the reviewers for their careful reading of the manuscript and constructive comments. We believe that their questions and comments have allowed us to improve the manuscript. Below, we reply to their individual remarks.

Reviewer #1

The manuscript by Tejero et al. reports the cryo-EM structures of the jumping spider rhodopsin isoform-1 (JSR1) in complex with the human Gi and chimeric jsGiq heterotrimers. These structures provide insights into different activation mechanisms for the monostable and bistable opsins. However, one major concern is that the authors do not provide enough complementary data, such as receptor mutagenesis, binding and signaling data, to support their findings.

We appreciate that the manuscript does not sufficiently emphasize our previously published work on the characterization of JSR1 mutants (**Nagata et al, *Commun Biology* (2019)**, reference [12]). To address this, we made a table below listing a number of JSR1 mutants and their functional effects (measured by spectroscopic analysis) that we have characterized in this previous publication. Also, key residues stabilizing retinal have been discussed before in the paper reporting the dark-state JSR1 crystal structure (**Varma et al, *PNAS* (2019)**, reference [11]).

Mutant	Effect
E194^{45.44}Q (distal counterion)	Dramatic decrease in the Schiff base pKa, indicating that Glu194 serves as the Schiff base counterion.
E194^{45.44}D (distal counterion)	Blue shift of 8 nm in λ_{max} compared to WT, meaning the Schiff base is less protonated in E194D than WT due to the carboxyl group of E194D being further away from the Schiff base. This description is supported by the dark-state crystal structure (2019 Varma et al, 6i9k.pdb)
Y126^{3.28}F ("proximal counterion")	Did not show any obvious decrease in the Schiff base pKa compared to WT, implying that Tyr126 ^{3.28} (at the position of the proximal counterion in bleachable opsins) has a minimal role in stabilization.
S199^{45.49}A (ECL2)	Substantial decrease in the Schiff base pKa, suggesting that Ser199 participates in stabilization of the protonated Schiff base.
S199^{45.49}C (ECL2)	Decrease in the Schiff base pKa to a value of 8.5, indicating the importance of Ser199's polar nature.
S199^{45.49}T (ECL2)	Retained higher Schiff base pKa values, suggesting the polar character at this position is crucial for protonated Schiff base stabilization.
S199^{45.49}N (ECL2)	Displayed relatively high Schiff base pKa values, suggesting additional structural features may assist in raising the Schiff base pKa.
S199^{45.49}F (ECL2)	Caused a substantial structural change around the Schiff base, shifting λ_{max} to the UV region, but did not affect the properties of the photoproduct.

These mutagenesis data suggested, for instance, that residue S199 is involved in a hydrogen bond network in the ground dark state but not in the active illuminated state, which is fully supporting our structural findings. We are currently carrying out an extensive mutagenesis study of the retinal binding site and activation switches of JSR1. We expect to generate a richer set of functional data that allows us to advance in our understanding of bistability. However,

in our opinion, adding data on a few of these mutants to the present manuscript would currently not help to clarify the mechanism further and decisively, and we plan to complete this work and publish separately.

The revised manuscript now contains multiple references to the published mutagenesis data. Amongst others, the following references to mutagenesis data are used to support our findings:

Page 8, line 240: “In a previous work [12], we have shown that mutating Tyr126^{3.28} to phenylalanine results in no change in the Schiff base pKa...”

Page 8, line 248: “These observations are consistent with published mutagenesis data suggesting that Glu194^{45.44} acts as the counterion in the active state, but that Ser199^{45.49} is not involved in the hydrogen bond network linking the PSB to the Glu194^{45.44} [12].”

Page 15, line 428-431: “As Glu194^{45.44} is known to retain its role as counterion in both inactive and active states [12]...”

Page 15, line 440: “This observation is consistent with our mutagenesis data showing that altering the polar nature of Ser199^{45.49} lowers the pKa of the Schiff base in the inactive state [12].”

In addition, in previous works we have also characterized the photocycle of JSR1 using FTIR, resonance Raman, and time-resolved UV-Vis spectroscopy (**Ehrenberg et al, *Biophys J* (2019)**, reference [16]; **Hanai et al, *Biochemistry* (2023)**, reference [42]). This allowed us, for instance, to obtain kinetic details of the rise and decay of the photocycle intermediates and elucidate structural changes of the retinal chromophore upon illumination. In particular, these data indicated that water molecules close to the Schiff base play an important role in the activation of JSR1, which also support our structural data:

*Page 15, line 431: “The model is supported by resonance Raman experiments showing the presence of a water molecule that hydrogen bonds to the PSB in both the inactive 9-cis and active all-trans states (**Ehrenberg et al, *Biophys J* (2019)**).”*

*Page 15, line 433: While the inactive state model is based on the crystal structure of JSR1 bound to 9-cis retinal, Fourier transform infrared spectroscopy analysis suggests that the hydrogen bond network around the retinal is similar when the endogenous chromophore, 11-cis retinal, is bound (**Hanai et al, *Biochemistry* (2023)**).*

Our mutagenesis and structural studies have already identified that the S199F mutation leads to deprotonation of retinal in the ground state of JSR1, leading to a complete spectral separation between dark and illuminated states (page 17, line 517-520). This is a key feature for potential optogenetic applications (see, for instance, **Hagio et al, *eLife* (2023)**, reference [45]; page 17, line 519), and we are exploring this possibility in a follow up work.

In the accompanying paper (<https://www.pnas.org/doi/10.1073/pnas.2406814121>), we performed extensive characterization of binding of the retinal 6.11 analogues to JSR1 and performed a thorough spectroscopic characterization. In addition, we demonstrated signaling

via human Gi and Gq proteins *in vitro* and complex formation with both G protein heterotrimers. This paper is now cited as **reference [18]** in this manuscript.

Our published complementary data support our interpretation of the structural findings, and we believe that this is now better reflected in the manuscript.

Furthermore, a few points should be addressed:

1. The authors used a human Gi/jumping spider Gq chimera (jsGiq), which was generated by engineering the hGi to match the AA sequence of the jumping spider visual Gq1 in the receptor binding interface. However, this chimeric G protein is not biological relevant. Have the authors tried the jumping spider Gq instead of the chimera?

We did indeed first attempt to use the jumping spider visual Gq1 protein, expecting that a ‘native’ G protein would yield a more stable complex with JSR1. However, we could not identify conditions under which jumping spider Gq1 was purified for structural studies. This is consistent with the known difficulties of working with G proteins of the Gq subtype *in vitro*. We were therefore unable to form a complex between JSR1 and jumping spider Gq1 suitable for structural determination.

We then decided to create a chimera that included the C-terminal sequence from spider visual Gq1 in the ‘core’ of a more amenable G protein. Similar strategies have been used to solve other GPCR/G protein complexes (Xia *et al*, *Nat Comm* (2021); Duan *et al*, *Nat Comm* (2022)). In our laboratory, we have created a native human Gi construct which works very well *in vitro* and that we have used in other structural studies (Tsai *et al*, *elife* (2019, reference [44]); Isaikina *et al*, *Sci Adv* (2021)). We have also shown that JSR1 activates human Gi and Gq/11-like proteins *in vitro* (Rodrigues *et al*, *PNAS* (2024), reference [18]). Thus, in this work, we decided to use our Gi construct as a template to generate the jsGiq chimera.

We have modified the manuscript to clarify this point:

Page 5, line 146: “Additionally, we attempted to use the jumping spider visual G_q (jsG_q) to form the complex. However, production of active and pure jsG_q for structural studies was not successful. Therefore, we decided to create a human G_i/jumping spider G_q chimera (jsG_{iq}) by engineering hG_i to match the sequence of jsG_q at the interface where the hG_i protein contacts the receptor based on our JSR1-hG_i cryo-EM structure.”

2. For the JSR1-hGi and JSR1-jsGiq cryo-EM data processing, the authors used different pipeline softwares, cryoSPARC for JSR1-jsGiq and Relion for JSR1-hGi. Have the authors tried to use the same software for data processing? As to my concern, Relion and cryoSPARC may give different results in some cases. So, it is worthwhile to check if one or the other software may give better result.

The JSR1-hGi complex sample was challenging to form and purify, and this unsurprisingly caused difficulties during cryo-EM data processing. On the cryo-EM grids, only a small fraction of the particles corresponded to the complex. Thus, a large amount of data (~8.4

million particles) had to be collected and multiple processing strategies explored, until progress could no longer be made. We processed both JSR1-hGi and JSR1-jsGiq datasets exploring both Relion and cryoSPARC at different stages of the data analysis. We observed that the JSR1-hGi dataset processed by the described Relion pipeline resulted in a better map, while for the JSR1-jsGiq sample the described cryoSPARC pipeline yielded better results.

Meanwhile, it was noticed that the particles in the JSR1-hGi dataset were polished in the final refinement, while not in the JSR1-jsGiq dataset. Why is that? If the authors did not polish the particles in the JSR1-jsGiq dataset, this should be done, as this process usually increases map resolution and quality.

The JSR1-hGi data set was polished (as a logical but computationally very demanding step in the standard Relion-pipeline processing). However, the outcome was not significantly improved, as we can see that the data itself was not holding promise for obtaining higher resolution. As to the JSR1-jsGiq data set, we have performed local CTF correction for each particle, but no further polishing could be done due to heavy computational requirements of working with the the large number of micrographs.

3. In the JSR1-jsGiq dataset, more than 74k images were collected. And a total number of 8.4 million particles were picked and extracted for later processing. The authors obtained two final maps with 134k and 159k particles. It is not unexpected to have many particles discarded, but for full clarity, the authors should modify the workflow in Supplementary Figure 5 by displaying more information and intermediate maps. Also, angular distribution plot should also be included. The same should be applied to the JSR1-hGi dataset in Supplementary Figure 3.

There seems to be some misunderstanding. The JSR1-jsGiq dataset has 74k images, giving 3.2 million picked particles.

We have modified both Supplementary Figure 3 and 5 according to the suggestion of the reviewer to include angular distribution plots.

Minor:

1. Lines 230-242, Glu19445.54 should be Glu19445.44, as Glu19445.44 is used for the numbering scheme in the rest of the paper, including P15, Fig. 2, Fig. 6, Supplementary Fig. 6-7.

We have corrected these typographical mistakes, and we thank the reviewer for their sharp eye.

2. Throughout the text, a subscript should be used for G protein subtype expression, e.g. G_q instead of Gq.

Thank you for pointing this out. We have now corrected this throughout the manuscript.

Reviewer #3

The manuscript entitled "Active state structures of a bistable visual opsin bound to G proteins" by Tejero et al. gives the first insight into the structure of the active state of bistable rhodopsin and its formation and suggests a potential G protein selectivity mechanism.

Animal rhodopsins are G protein-coupled receptors and photo-sensitive pigments that contain an 11-cis retinal chromophore. Upon light absorption, isomerization of 11-cis retinal to its all-trans form induces protein conformational changes that activate the GPCR and allow signal transmission to the G protein. Light-activated vertebrate rhodopsin loses the all-trans-retinal due to hydrolysis of the Schiff base linkage and requires binding of 11-cis retinal to restore rhodopsin. In contrast, invertebrate rhodopsins do not lose the retinal. They rather absorb a second photon to isomerize the retinal back to its initial 11-cis state and are therefore called bistable rhodopsins, i.e. stable in the inactive and active state. So far it was not possible to obtain a structure of invertebrate rhodopsin in the active state, because of the largely overlapping absorption profiles of the inactive and active states that typically cause a mixture of these states. The authors Tejero et al. used two approaches to obtain complexes between jumping spider rhodopsin 1 (JSR1) and G protein. Reconstituting JSR1 with 9-cis retinal caused a 30 nm spectral blue shift, which allowed specific illumination to enrich all-trans-retinal containing active JSR1 that could be complexed with the human inhibitory G protein (hGi) for structure analysis by cryo-electron microscopy (cryo-EM). In a second approach JSR1 was reconstituted from the opsin apo protein and a retinal analog, i.e. all-trans-retinal with a 6-carbon ring locking the C11-C12 double bond in the trans configuration (ATR6.11). In contrast to all-trans-retinal, ATR6.11 did form a stable pigment with opsin, which is capable to activate G protein and form a complex with a human Gi/jumping spider Gq chimera (jsGi_q) for cryo-EM analysis.

The authors present a cryo-EM map of JSR1-hGi complex (4.9 Angstrom resolution) and two maps of JSR1-jsGi_q complexes (4.06 and 4.17 Angstrom resolution), giving first insight into commonalities and differences of the activation mechanisms of monostable and bistable rhodopsins. The discovery of ATR6.11 as an agonist suitable for JSR1-G protein complex formation is an important step towards understanding the active structure of JSR1 and has big potential to also be useful for other bistable rhodopsins. Bistable rhodopsins are not only employed for vision in invertebrates, but also serve in non-visual light perception in vertebrates, and therefore the elucidation of their activation mechanism is of great interest in this field.

The present work, however, has a couple of points which need clarification and/or additional experiments:

1) The retinal analog ATR6.11 is a key element for the present work. Its biochemical and biophysical characterization as chromophore for JSR1 has been reported in a BioRxiv preprint (Ref. 18). It is important that this characterization is either published separately or as part of the present manuscript.

We agree with the reviewer, the manuscript presenting the characterization of ATR6.11 is now published in PNAS (<https://www.pnas.org/doi/10.1073/pnas.2406814121>).

2) The authors state that the JSR1(ATR6.11)-jsGiq complex sample for cryo-EM study was purified under ambient light conditions. According to the method section, the final SEC purification was done with a SRT-C 300 column for JSR1(ATR6.11)-jsGiq (Supplementary Fig. 1C) and a Superdex 200 increase column for JSR1-hGi (Supplementary Fig. 1A). The SEC profile of JSR1(ATR6.11)-jsGiq doesn't show well separation of the complex. Peaks 1 and 2 are barely separated and contain the same proteins. Is this an effect of the different columns used? Or is it due to purification under ambient light? Does the non-single peak reflect the two structures observed in cryo-EM structure determination? Where is the unbound G protein peak? Why was not a Superdex 200 column used? These questions arise because Figure 2 of the BioRxiv paper shows good separation for both, the JSR1(ATR6,11)-hGi complex and the JSR1(ATR6.11)-human Gq complex.

Thank you for raising up these questions. First, we would like to talk about the impact of light on the SEC purification profile. The figure below shows the SEC profiles of JSR1•ATR6.11 under dark or light condition mixed with either human Gi (upper panel) or human Gq (lower panel). The SEC profiles of samples under light and dark conditions are basically identical. Therefore, we rule out an impact of light on dissociation of the complex of JSR1•ATR6.11.

Sepax Zenix-C SEC 300
7.8x300 column
1CV = 14.34 mL

As for the peak separation in the JSR1-hGi and the JSR1-jsGiq samples shown in supplementary Figure 1A/C, here are the different parameters between these two sample runs that might influence the SEC results: 1) G protein, 2) detergent, 3) SEC column, and 4) injected protein quantity. Here we discuss those differences and how they impact the SEC results:

- 1) G protein: during EM data processing, we observed that JSR1-hGi is prone to dissociate, while JSR1-jsGiq stays intact. We also see that JSR1 forms more contact interface for binding jsGiq than for binding hGi in our structural analysis, suggesting tighter binding to jsGiq than hGi. This is also coherent to the SEC profiles of JSR1-hGi and JSR1-jsGiq, where peaks of free JSR1 and free G protein are observed in the JSR1-hGi sample but not in JSR1-jsGiq sample. All these observations point towards that JSR1 forms a tighter and more stable complex with jsGiq than with hGi. Therefore, it explains that there are no unbound G protein peak and JSR1 peak coexisting in the JSR1-jsGiq sample. Of course, when the JSR1/jsGiq molar ratio is far off from 1, a peak of free JSR1 or free jsGiq appearing in the SEC profile is expected.
- 2) JSR1-hGi was purified in LMNG, and JSR1-jsGiq in DDM. Although detergent can shift the resolution of a SEC column, we expect the differences between LMNG and DDM in the SEC purification is rather small because both LMNG and DDM detergents have similar micelle sizes and properties, and also only very low concentration was used (LMNG only present in the injected JSR1-hGi sample and not in the SEC buffer; DDM used at 0.01% concentration in the SEC buffer).
- 3) Due to the set-ups at that time in our lab, we used two different SEC columns: JSR1-Superdex200 Increase 10/300, Sepharose based for JSR1-hGi, and SRT-C 300 10/300, silica based for JSR1-jsGiq. They have the same column volume (24 mL) but different resin bead material, and therefore it is difficult to disentangle the effect of all of these variables on the final SEC profiles. Besides, the peaks of JSR1-hGi and of JSR1 are 1.06 mL apart (supp. Fig. 1A). However, the peak 1 and the peak 2 of JSR1-jsGiq are only 0.47 mL apart (supp. Fig. 1C). These two peaks are much closer and therefore worse separated.
- 4) Injected protein quantity can greatly influence the separation of the protein populations. As the injected JSR1-jsGiq sample has higher quantity than the JSR1-hGi sample, which also resulted in the peak 280 nm absorption in JSR1-jsGiq 3.5-fold higher than that in JSR1-hGi, it is expected that JSR1-jsGiq SEC profile shows worse separation than JSR1-hGi.

For the JSR1-jsGiq sample, we combined both peak 1 and peak 2, concentrated the combined fractions, followed by flash freezing on EM grids. Therefore, the EM data acquired contains complex assemblies from both peak1 and peak 2. As the reviewer has mentioned, each peak may preserve a particular conformation, we however could not directly prove it based on the data we currently have. Even if specific conformations can be biochemically purified, the variability analysis of the EM data shows a spectrum of conformations. Therefore, we doubt that isolated populations possess static conformations and instead consider that each population contains a dynamic spectrum of protein conformations.

These SEC profiles in the newly provided figure above were recorded using the analytical column, Sepax Zenix-C SEC 300 7.8/300, 1CV = 14.34 mL. This column was also used for the SEC experiment in the PNAS paper (<https://www.pnas.org/doi/10.1073/pnas.2406814121>).

We believe that JSR1-hGi and JSR1-hGq show better separation in the accompanying manuscript because (1) a lower concentration of injected protein samples was used, and (2) JSR1 is more prone to dissociate from hGi and hGq than from jsGiq. For the structure paper, we had to use wider columns (10mm vs. 7.8 mm wide) and inject higher protein quantities for large-scale preparative protein production.

We have modified the manuscript to include the following information:

In Supplementary Figure 1A and C: The fraction range collected for freezing EM grids and the retention volume of the peaks are now marked in the figure.

Column size is added to the Methods section (page 21, line 657 and 664).

It is very confusing that the sample preparation has not been performed in the dark. Even if the retinal is chemically locked at the C11-C12 double bond, there is still the possibility that ambient light may cause trans-cis isomerization of other bonds of ATR6.11 in JSR1.

We see that our attempt to describe the complex preparation for cryo-EM concisely in the results could reasonably lead to confusion and have updated the description in the Methods section to clarify the issue.

The light sensitive component of the complex, JSR1-ATR6.11 was prepared and purified completely under dim red-light conditions, as described in the method section “JSR1 Expression and Purification”. Furthermore, the one-hour incubation of JSR1-ATR6.11 incubation with jsGiq and apyrase to induce complex formation and sample concentration prior to size exclusion chromatography were also performed under dim red-light conditions.

Due to the practicalities of performing size exclusion chromatography at 4°C, the sample was briefly exposed to ambient light during injection into our liquid chromatography system and during fraction collection. Due to the opaque nature of the HPLC column casing, the sample was protected from light while on the column. We have updated the methods sections to reflect this (page 21, line 654-665).

In our accompanying manuscript we describe a spectroscopic shift in the retinal absorption peak following relatively prolonged (60 seconds) illumination under bright 519 nm light (1.0 mW cm⁻²), which indicates that it is unlikely that a brief exposure to ambient light will yield a detectable fraction of cis-trans isomerization. The bright 519 nm illumination yielded a photoproduct with greater agonist activity than the originally reconstituted ATR6.11. Despite efforts to analyse retinal 6.11 oximes extracted from JSR1 by HPLC, we are as yet unable to conclusively determine the molecular bases of the photoreaction.

Our cryo-EM dataset will only select for active JSR1 as this is a condition for complex formation and only complexes are selected during 2D classification. We know that the C11=C12 bond is locked and therefore may not isomerize. We have re-inspected the cryo-EM maps and also concluded that the 9-cis and 13-cis isomers cannot be fitted to the density well. This suggests that if there is receptor present in these states, they represent a negligible fraction of the receptor population selected for 3D reconstruction of the complex.

Based on the data, we are unable to exclude that a photo-induced syn/anti isomerization of the Schiff base may occur. The resolution of our data is not sufficient to distinguish between

syn/anti Schiff base isomers and both may be fitted with good agreement to the experimental map. We chose to model the anti-isomer, because both the dark state and photoproduct retain a protonated Schiff base (Figure 1G, accompanying manuscript). The anti-isomer is compatible with the map and would allow the protonated Schiff base to be stabilized by the Tyr126^{3,28} hydrogen bond network, while it is unclear how the PSB would be stabilized in a syn conformation.

In conclusion, while it was, unfortunately, not possible to complete all steps of sample preparation under dim light conditions, exposure to ambient light was minimized. Based on our interpretation of the structural data, the dominant isomer of retinal 6.11 present in the JSR1-jsGiq complex is all-trans. Identifying the nature of any photoproducts will require detailed study using vibrational spectroscopy and retinal extraction, which we consider to be a further research project.

We therefore update the Method section to include the sample illumination for both samples (Page 21, line 659-663).

Active cis retinal forms have been described for locked retinal reconstituted bovine rhodopsin and have been discussed by the authors in the BioRxiv preprint (Ref. 9, Gulati et al. 2017 PNAS). The complexes of JSR1(ATR6.11) with hGi and hGq, respectively, in the BioRxiv preprint have been purified in the dark and are well separated by SEC. Good separation is also seen for the JSR1-hGi complex in the present study, which was obtained by illumination of JSR1(9-cis retinal) followed by complex formation with hGi and purification in the dark. Perhaps the light conditions and Superdex column cause the difference for JSR1(ATR6.11) and should be clarified.

Thanks for the comments. The point of illumination has been addressed in a reply above, and we rule out this possibility. We attribute the change in the separation of the peaks to the injected quantity, the columns used, and the differences in the sample. Therefore, we added details of the SEC columns used to the legend of Supplementary Figure 1.

3) The BioRxiv preprint shows that ATR6.11 acts as an agonist for JSR1. The activity of JSR1(ATR6.11) is increased when the pigment is illuminated. Given that cis isomerizations of 11-cis locked retinal in bovine rhodopsin cause activity, it would be interesting to know the retinal composition upon illumination of JSR1(ATR6.11). A mixture of isomers could be an explanation for the result of the JSR1(ATR6.11)-jsGiq complex purified under ambient light.

We agree with the reviewer that it would be interesting to know the retinal composition of JSR1(ATR6.11) after intense illumination and as a result of this comment we have initiated experiments where we extract the analogs from JSR1 with hydroxylamine. We have subjected the extracted retinal oximes to normal phase HPLC and observe formation of a new peak with a shifted retention time. While it is clear that there is a light induced change in the retinal composition, it is not possible to assign which retinal isomer(s) are formed after illumination. The cited paper by the Palczewski group used NMR to determine the absolute configuration of the retinal analog photoproducts; however, we are unable to do this as the HPLC peaks of the photoproduct and ATR6.11 oximes overlap and are not separated under our current experimental conditions.

We are however able to exclude the formation of a di-cis retinal based on the shape of the density in the EM maps. We are aided in this interpretation by the ring constraining the 11-bond in a trans configuration, which limits the number of possible cis isomers that may form.

4) The lower resolution of JSR1-hGi complex was explained by a possible mixture of retinal isomers and the reason to use ATR6.11 in the JSR1(ATR6.11)-jsGiq complex. The authors also suggested other factors such as heterogeneity of the complex and the tendency of dissociation for the JSR1-hGi complex. A mixture of all-trans-retinal and 11-cis retinal should yield a mixture of complex and dissociated complex, which particle picking in cryo-EM analysis should be able to separate. It might be possible that the higher affinity of jsGiq might allow stable complex formation. Did the authors test this?

Retinal heterogeneity was reported for JSR1 reconstituted with 9-cis retinal followed by illumination (**Ehrenberg et al, *Biophys J* (2019)**, reference [16]) and this problem can be solved by using ATR6.11. In the beginning we also thought that ATR6.11 alone could improve the resolution of the JSR1-hGi complex, but it was not the case. Prior to design of the jsGiq chimera, we also made the JSR1•ATR6.11-hGi complex and performed cryo-EM analysis. However, the complex still dissociated, indicating that the complex stability was not much improved by merely changing retinal to the retinal analog. Instead, we are convinced that the key factor in complex stability is the G protein. As revealed in our structural analysis, jsGiq has more atomic interactions to JSR1 than hGi does and better resolution. At the stage of 2D classification during EM data processing, we could already observe that the number of 2D classes containing either only JSR1 in detergent micelle or only G protein heterotrimer was much lower when using jsGiq. This is a proof that jsGiq indeed forms a more stable complex with JSR1 than hGi. However, the affinity between GPCR and G proteins is difficult to quantify (**Wu et al., *Proc National Acad Sci* 118, e2024146118 (2021)**). Still, we observe that dissociation of the complex is very prominent in the hGi data and not in jsGiq data.

Until now, we have not tested complex formation between jsGiq and illuminated JSR1•9-cis retinal (to generate the all-trans species). While this is a reasonable suggestion, there are too many combinations of parameters that can be adjusted in the hope of obtaining a higher resolution dataset for all of them to be tested.

If there is heterogeneity, it may arise from other cis isomers different from 11-cis. Supplementary Figure 6CD shows differences in the density maps for ATR6.11 and all-trans-retinal when present in JSR1. Especially striking is the difference around the C13-C14 bond. The retinal in JSR1-hGi (blue mesh) may contain a mixture of retinals including one with a 13-cis bond, perhaps as a result of the illumination conditions used for 9-cis retinal reconstituted JSR1. Unfortunately, the figure legends lack in several cases clarity about the sample, which needs to be corrected. If 13-cis is present, it would be similar to a potential trans-cis isomerization of ATR6.11. Such a 13-cis isomer could be important as it may have a functional role and requires further investigation.

HPLC analysis of retinal photoproducts generated by JSR1•9-cis retinal illumination reported by Ehrenberg et al (**Ehrenberg et al, *Biophys J* (2019)**, reference [16]) has shown that the retinal isomers generated are 73% all-trans, 19% 11-cis, 5% 9-cis, and only 3% for 13-cis retinal. Based on this statistic, we think it is quite unlikely that JSR1•13-cis retinal is present in a sufficient proportion to contribute significantly to the final map.

JSR1•9-cis and JSR1•11-cis retinal are inactive, and therefore do not form complexes with the G protein. We therefore expect that the receptor bound to these isomers is primarily present in the free JSR1 peak in the size exclusion chromatogram (elution volume=11.95 mL, supp. Figure 1A), these fractions were not used for cryo-EM grid preparation. Furthermore, dissociated receptors are excluded from the final cryo-EM during 2D classification step. Since we strongly focus in selecting particles of the JSR1-hGi complex, it also means that we select the conformation favoring G protein binding.

We agree with the reviewer that it is important to consider the possibility that other retinal isomers contribute to our density maps. In response to the suggestion, we attempted to fit 13-cis retinal 6.11 into our jsGiq density maps, however, we could not find a pose consistent with the maps.

We have sought to improve the clarity of the figure legends. However, if there are any further specific improvements that can be made, we are open to doing so.

On page 7, the authors mention that cryo-EM densities of JSR1-hGi and JSR1-jsGiq are similar in pose but there is a slight difference in lysine side chain and the Schiff base. For readers who are interested in residue changes upon activation, however, the different positions of Schiff base wouldn't be trivial. A discussion of the different positions of standard retinal and ATR6.11 Schiff base should be added for clarification. A 13-cis isomer might potentially be an explanation.

As discussed above, due to the HPLC analysis of JSR1 photoproducts, we consider the presence of 13-cis retinal to be unlikely in any significant quantity.

We agree with the reviewer that the Schiff base and lysine conformations are important for understanding the activation of JSR1. Due to the resolution of our data, we aim to gain as many structural insights as possible, without over interpreting the differences. For this reason, we have primarily focused on the JSR1-jsGiq, which was solved at higher resolution and for which the side chain and retinal conformations can be modelled with greater confidence. We have added a sentence to clarify this reasoning to the reader:

Page 7, line 205-208: "These structural differences may reflect the ambiguity of modelling flexible side chain conformations at this resolution, here we focus on the JSR1-jsGiq_1 structure, which was solved at higher resolution and for which the side chain and retinal conformations could be modelled with greater confidence."